# Generative Large Neighborhood Search: Scalable Set Cover Optimization via Discrete Diffusion

Achref Jaziri [1] [*]   Thibaut Cuvelier [2]   Bruno De Backer [2]

## Abstract

Large-scale Set Cover Problems (SCP) with millions of variables and complex cost structures require high-quality solutions within seconds, yet remain beyond the reach of exact solvers and pose severe generalization challenges for neural methods. Such problems necessitate decomposition into bounded subproblems; however, when the induced subproblem topology differs from that observed during training, existing neural approaches often fail to transfer reliably. We introduce Generative Large Neighborhood Search (GLNS), which reframes neighborhood selection as generation using a discrete diffusion model. Our key insight is that the diffusion denoising trajectory exposes variables exhibiting high prediction instability across timesteps and identifies regions where local repair yields downstream improvement. GLNS exploits this trajectory-level signal to construct high-impact neighborhoods via a localized, bounded-complexity generative sampling procedure, enabling robust neighborhood selection without retraining. As a result, GLNS transfers effectively across cost regimes and instance scales within SCP. Under tight and equal wall-clock budgets, GLNS consistently outperforms established neural baselines and achieves competitive performance with state-of-the-art heuristic solvers. These results demonstrate trajectory-guided generation as a scalable framework for large-scale SCP and suggest potential relevance to other constrained optimization settings.

## 1. Introduction

The Set Cover Problem (SCP) is a core problem in combinatorial optimization (CO) (Chvatal, 1979). Despite decades of study, the computational scaling of the static offline problem remains a critical operational bottleneck. Industrial instances frequently span $10^5$ to $10^9$ variables, a regime where the exponential dimensionality of the discrete search tree renders exact Mixed-Integer Programming solvers computationally prohibitive and where classical heuristics show clear limits in solution quality.

Despite its practical relevance, SCP has received little attention in neural CO benchmarks. On small instances, exact solvers leave little room for neural methods to add value; on large instances, exact solvers face memory and runtime limits, while neural approaches struggle with feasibility and generalization. This gap makes SCP a natural benchmark for hybrid neural–symbolic methods in regimes where classical approaches break down.

Scaling Set Cover to large instances requires decomposition into bounded subproblems that fit within practical memory and runtime constraints. However, decomposition fundamentally alters the learning problem. Subproblems extracted from large instances can exhibit topology, cost-distributions, and constraint densities that differ substantially from those seen during training. As a result, neural methods trained on fixed-scale instances encounter a pronounced distribution shift at deployment. Approaches that rely on point predictions—mapping local features directly to repair decisions—tend to overfit to training-scale statistics, which helps explain the well-documented difficulty of neural CO methods in generalizing across scales (Joshi et al., 2020; Fu et al., 2021).

Existing neural approaches to Large Neighborhood Search (LNS) typically frame neighborhood selection as a deterministic prediction task: a model is trained to classify which variables should be modified or to rank repair candidates based on learned features. However, this formulation discards valuable information about the uncertainty and interdependence structure of the solution space. In SCP, effective repairs often require coordinated changes across groups of structurally coupled variables, sets that share many elements or participate in overlapping constraints. Point predictions

---

*Work performed during an internship at Google Research
[1]Goethe University, Frankfurt, Germany [2]Google Research, Paris, France. Correspondence to: Achref Jaziri <jaziri@em.uni-frankfurt.de>, Thibaut Cuvelier <tcuvelier@google.com>.

*Proceedings of the 43rd International Conference on Machine Learning*, Seoul, South Korea. PMLR 306, 2026. Copyright 2026 by the author(s).

that score variables independently cannot capture this relational structure, leading to myopic selections that fail when problem characteristics shift.

We propose Generative Large Neighborhood Search (GLNS), a framework that reframes neighborhood selection as a generative process guided by discrete diffusion models. Our key insight is that the iterative denoising trajectory exposes structural coupling through prediction instability: variables whose cluster assignments fluctuate across diffusion timesteps tend to be high-value candidates for joint repair. Unlike methods that rely solely on final model outputs, GLNS extracts trajectory sensitivity scores that measure prediction variance across the denoising sequence, providing a learned heuristic signal for identifying coherent repair regions. Critically, GLNS achieves computational scalability by combining this neural guidance with problem-specific decomposition: localized subproblems (halos) are extracted around high-cost regions and passed to the diffusion model, ensuring constant GPU memory regardless of global instance size.

This work makes three main contributions to neural combinatorial optimization. First, we introduce a diffusion-guided LNS framework that exploits denoising trajectory dynamics rather than final model predictions for neighborhood selection, showing empirically that temporal prediction variability can act as a reliable proxy for structural importance in discrete optimization problems. Second, we propose a halo-based decomposition strategy that bounds neural computation while allowing scaling to larger instances, enabling scalable deployment on problem instances with orders-of-magnitude larger variable counts without retraining. Third, through extensive experiments on standard benchmarks, we demonstrate that GLNS transfers robustly across problem scales and cost distributions, consistently outperforming discriminative neural baselines and remaining effective in large-scale regimes where exact solvers struggle to produce high-quality feasible solutions within practical time limits. Together, these results position trajectory-guided neighborhood selection as a practical approach for integrating learned guidance into large-scale Set Cover optimization pipelines. We view GLNS as a complementary component in hybrid optimization pipelines, providing robust anytime improvements and strong transfer in large-scale regimes where exact solvers struggle under tight time budgets.

## 2. Background

We consider a global optimization problem defined on a bipartite graph $\mathcal{G} = (\mathcal{U}, \mathcal{C}, \mathcal{E})$, where $\mathcal{U}$ denotes variables (sets) and $\mathcal{C}$ denotes constraints (elements). The objective is to find a binary assignment $\mathbf{x} \in \{0, 1\}^{|\mathcal{U}|}$ minimizing a linear cost $J(\mathbf{x}) = \mathbf{c}^\top \mathbf{x}$ subject to covering constraints $\mathbf{Ax} \geq \mathbf{1}$. We focus on improving an incumbent feasible

solution by learning the *structure of repairs* rather than constructing solutions from scratch.

**Neural CO and LNS.** Neural CO approaches have shown strong performance on routing and sequencing problems such as TSP and VRP (Bello et al., 2016; Kool et al., 2018). However, Set Cover (SCP) remains challenging due to its global overlap structure. Unlike single-graph problems (e.g., Maximum Independent Set) where violated constraints implicate a fixed local adjacency, dropping a set in SCP can uncover elements anywhere in the instance, demanding non-local repairs driven by global topology (Karalias & Loukas, 2020; Alon & Yahav, 2020). Furthermore, while inductive biases have enabled scale generalization in routing (Fu et al., 2021; Kool et al., 2018), these mechanisms do not directly transfer to bipartite covering problems where both set and element dimensions grow (Joshi et al., 2020). This motivates hybrid neural–symbolic approaches that guide classical optimization procedures rather than directly predicting solutions (Gasse et al., 2019; Wu et al., 2021; Li et al., 2025).

Large Neighborhood Search (LNS) (Shaw, 1998) provides a natural framework for such hybridization by iteratively destroying and repairing subsets of variables within a feasible solution. Prior neural LNS methods typically model neighborhood selection via classification, ranking, or reinforcement learning (Khalil et al., 2016; Song et al., 2020; Hottung & Tierney, 2019; Wu et al., 2021), relying on point-estimate predictions to guide repair.

**Diffusion Models for CO.** Recent work has applied diffusion models to CO (Sun & Yang, 2023; Sanokowski et al., 2024; Ma et al., 2025; Li et al., 2024; 2023), including diffusion-guided LNS approaches that sample neighborhoods from the final denoised output (Feng et al., 2024). While effective in capturing global structure, these methods typically treat the diffusion process as a black box, discarding information contained in intermediate inference dynamics.

In contrast, our approach leverages the diffusion denoising trajectory as a heuristic signal for neighborhood selection. Prior work has shown that intermediate diffusion states can provide meaningful uncertainty estimates (Graikos et al., 2022), and that prediction instability can highlight structurally important components (Toneva et al., 2018; Liu et al., 2020). We adapt this perspective to discrete optimization by using prediction variability across timesteps to identify variables that tend to participate in tightly coupled constraints, making them effective candidates for joint repair. Combined with problem-specific decomposition that curbs computational complexity, this trajectory-driven approach enables scalable optimization in regimes where existing neural methods struggle and exact solvers exceed practical limits.

# 3. GLNS: Trajectory-Guided Large Neighborhood Search

Scaling neural LNS to million-variable instances requires addressing two challenges: bounding computational complexity and maintaining robustness under distribution shift. GLNS addresses both through a unified pipeline that combines *halo-based decomposition* with *trajectory-guided neighborhood selection*.

At a high level, GLNS iteratively extracts localized subproblems around high-cost regions, applies diffusion-based inference to identify groups of structurally coupled variables, and repairs each neighborhood using a symbolic solver. Rather than relying solely on the final diffusion output, neighborhood selection leverages prediction variability along the denoising trajectory, which empirically provides a transferable signal for identifying effective repair regions across varying problem structures.

GLNS proceeds in five stages: (i) halo-based subproblem extraction, (ii) state-change prediction via bipartite GNN, (iii) diffusion-based set clustering, (iv) trajectory-driven similarity construction, and (v) neighborhood partitioning with symbolic repair. Sections 3.1–3.5 detail each stage. See 4 for an overview diagram.

## 3.1. Halo-Based Decomposition

GLNS addresses the scale challenge by extracting localized neighborhoods—*halos*—around high-cost regions and optimizing each as an independent Set Cover instance. This ensures constant GPU memory and inference time regardless of global instance size, while preserving local structure that supports effective transfer across instance scales.

**Halo Seed Selection.** Effective destroy-and-repair moves target expensive sets that admit cheaper local replacements. We formalize this intuition via a *replaceability score*, defined as the ratio of a set's cost to the mean cost of its neighbors in the set-set sharing graph:

$$r(s) = \frac{c_s}{\bar{c}_{\mathcal{N}(s)} + \epsilon}, \quad \bar{c}_{\mathcal{N}(s)} = \frac{1}{|\mathcal{N}(s)|} \sum_{s' \in \mathcal{N}(s)} c_{s'} \quad (1)$$

where $\mathcal{N}(s)$ denotes sets sharing at least one element with $s$. Seeds are sampled from active sets proportionally to Boltzmann distribution over these scores:

$$p(s) \propto \exp(r(s)/\tau), \quad s \in \{i : x_i = 1\} \quad (2)$$

where $\tau$ controls exploration-exploitation tradeoff.

$k$-**Hop Expansion.** Starting from seeds $\mathcal{S}_0$, we expand through the bipartite set-element graph:

$$\mathcal{E}_t = \{e : \exists\, s \in \mathcal{S}_{t-1},\, A_{se} = 1\} \quad (3)$$
$$\mathcal{S}_t = \{s : \exists\, e \in \mathcal{E}_t,\, A_{se} = 1\} \quad (4)$$

After $k$ iterations, the halo is $\mathcal{H} = \bigcup_{t=0}^{k} \mathcal{S}_t$. For $k = 2$, this yields the 2-hop neighborhood in the set-set sharing graph. If $|\mathcal{H}| > N_{\max}$, non-seed sets are subsampled uniformly to bound complexity while preserving the high-replaceability.

Each halo induces a standalone SCP instance with incidence matrix $A_{\mathcal{H}}$ and cost vector $\mathbf{c}_{\mathcal{H}}$, which is passed to subsequent neural inference stages.

## 3.2. Predicting Variable Changes via Bipartite GNN

The search space of large-scale SCP instances is dominated by variables that remain unchanged in high-quality solutions. To focus downstream inference on regions with improvement potential, we first identify variables likely to differ between the current solution $\mathbf{x}_{\mathrm{curr}}$ and an improved local optimum. This prediction targets high recall, providing a broad candidate set that is subsequently refined by diffusion-based coordination.

**Bipartite Representation.** We represent each instance as a bipartite graph $\mathcal{G} = (\mathcal{U} \cup \mathcal{C}, \mathcal{E})$, where $\mathcal{U}$ denotes set variables and $\mathcal{C}$ denotes coverage constraints. Variable nodes $u \in \mathcal{U}$ are initialized with

$$\mathbf{h}_u^{(0)} = \left[ c_u,\ c_u/d_u,\ x_{u,\mathrm{curr}} \right],$$

encoding cost, cost efficiency, and current heuristic solution membership. Constraint nodes $c \in \mathcal{C}$ are initialized with

$$\mathbf{h}_c^{(0)} = \left[ d_c,\ \min_{u \in \mathcal{N}(c)} c_u,\ \mathrm{mean}_{u \in \mathcal{N}(c)} c_u \right],$$

encoding coverage density and local cost structure.

**Architecture.** We employ a bipartite GNN with directional message passing, parameterizing set-to-element and element-to-set interactions with distinct update functions to capture the asymmetric structure of coverage relations. A virtual context node aggregates global statistics and broadcasts them to all nodes, allowing local decisions to reflect system-wide cost distributions. Final representations pass through an MLP to produce per-variable change scores.

**Training Objective.** Variables requiring change are sparse, creating severe class imbalance. We optimize a composite loss combining Focal Loss (Lin et al., 2017), which down-weights easy negatives, with Tversky Loss (Salehi et al., 2017), which penalizes missed positives more heavily

than false alarms. This encourages the GNN to act as a high-recall filter: it identifies a superset of candidates with improvement potential, while diffusion provides the structural precision to group them into effective repair neighborhoods.

### 3.3. Diffusion-Based Structural Inference

The first-stage GNN provides pointwise estimates of sets that likely require modification, but it does not explicitly model which sets should be modified *jointly* to improve the solution. In practice, effective repairs often require coordinated changes across groups of structurally related sets. To capture this higher-order structure, we introduce a discrete diffusion model (Lou et al., 2023) that induces pairwise affinities among candidate sets. Crucially, we extract this structural signal not from the final denoised output alone, but from prediction dynamics along the denoising trajectory.

#### 3.3.1. CLUSTERING REPRESENTATION AND CONDITIONING

For an SCP instance with $m$ sets, the diffusion process models a symmetric binary matrix $X \in \{0,1\}^{m \times m}$, where $X_{ij} = 1$ indicates that sets $i$ and $j$ belong to the same repair cluster and should be modified together. The target matrix $X_0$ assigns all sets consistent with the incumbent solution to a single background cluster, while sets identified for repair are partitioned into finer clusters. This representation supports both trivial repairs (no modification) and structured multi-set repairs within a unified framework.

The diffusion model conditions on a graph $G_d = (\mathcal{S}, E)$ induced from the SCP instance, where nodes represent sets (with features: cost, coverage degree, element rarity, GNN prediction) and edges structural overlap weighted by shared elements. This conditioning graph provides the relational context required to infer meaningful repair groupings.

#### 3.3.2. FORWARD AND REVERSE DIFFUSION PROCESSES

At diffusion step $t$, the denoiser $p_\phi(X_0 \mid X_t, G_d, t)$ predicts the clean clustering matrix given a noisy observation $X_t$, the conditioning graph $G_d$, and the timestep $t$. We implement the denoiser as a graph attention network (Veličković et al., 2017).

We adopt a Bernoulli noise schedule. Let $\beta_t$ denote the noise rate at step $t$, $\alpha_t = 1 - \beta_t$, and $\bar{\alpha}_t = \prod_{i=1}^{t} \alpha_i$. The forward process corrupts the clean matrix $X_0$ according to

$$q(X_t \mid X_0) = \text{Bernoulli}\left(X_t; X_0\bar{\alpha}_t + \tfrac{1}{2}(1 - \bar{\alpha}_t)\right), \quad (5)$$

which replaces entries of $X_0$ with random binary values with probability $1 - \bar{\alpha}_t$. As $t$ increases, $X_t$ converges to a maximum-entropy distribution, erasing the initial structures.

The reverse process reconstructs latent similarity structure by sampling $X_{t-1}$ from $X_t$. Given a prediction $\hat{X}_0$, the reverse transition is defined entry-wise via

$$
\begin{aligned}
P(X_{t-1} = 1 \mid X_t, \hat{X}_0) &= \frac{\sqrt{\bar{\alpha}_{t-1}}\beta_t}{1 - \bar{\alpha}_t}\,\hat{X}_0 \\
&+ \frac{\sqrt{\alpha_t}(1 - \bar{\alpha}_{t-1})}{1 - \bar{\alpha}_t}\,X_t,
\end{aligned}
\quad (6)
$$

with probabilities clipped to $[0, 1]$ for numerical stability. This posterior corresponds to the standard Bernoulli reverse parameterization used in discrete diffusion models for binary variables, yielding a tractable entry-wise transition consistent with the underlying noising process (Austin et al., 2021; Hoogeboom et al., 2021). Repeating this procedure for $T$ steps yields a final similarity matrix $A_{\text{sim}}$ capturing latent structural dependencies among the selected sets.

#### 3.3.3. STRUCTURAL PRETRAINING

The space of valid clustering adjacency matrices forms a highly non-convex and combinatorial manifold. Directly optimizing repair quality from solver feedback is thus unstable. We first pretrain the diffusion model to generate structurally valid clusterings before refining it using solver feedback.

Pretraining minimizes a composite objective:

$$\mathcal{L}_{\text{pretrain}} = \mathcal{L}_{\text{recon}} + \lambda_{\text{sym}}\mathcal{L}_{\text{sym}} + \lambda_{\text{trans}}\mathcal{L}_{\text{trans}}, \quad (7)$$

where $\mathcal{L}_{\text{recon}}$ is a class-weighted binary cross-entropy loss, $\mathcal{L}_{\text{sym}}$ enforces symmetry, and $\mathcal{L}_{\text{trans}}$ encourages transitive consistency so that the predicted adjacency corresponds to a valid clustering. Rather than exhaustive $O(m^3)$ evaluation, we employ stochastic hard-triplet mining that focuses learning on the most critical transitivity violations while reducing complexity to $O(Km)$ (details in Appendix D.3.4).

#### 3.3.4. SOLVER-GUIDED REFINEMENT

Following structural pretraining, the model is refined using solver cost feedback. For each training instance, we sample a **candidate clustering partition $\pi$** derived from the model's predicted affinity matrix $\hat{X}_0$. This partition $\pi$ induces a repair move whose cost improvement $\Delta c(\pi)$ is measured. An advantage $A = \Delta c(\pi) - b$ is computed relative to a moving-average baseline $b$, and used to weight the denoising objective:

$$\mathcal{L}_{\text{refine}} = \mathbb{E}_{t, \hat{X}_0}\left[(b - \Delta c(\pi))\log p_\theta(\hat{X}_0 \mid X_t, G_d)\right] \quad (8)$$

Here, $\hat{X}_0$ represents the specific affinity matrix sampled from the reverse process that yielded partition $\pi$. This objective effectively acts as reward-weighted regression: it amplifies the likelihood of generating affinity matrices that result in high-quality discrete repairs ($\Delta c > b$) while suppressing those that fail to improve the solution.

## 3.4. Trajectory Sensitivity for Neighborhood Selection

The final diffusion output $\hat{X}_0$ reflects the model's converged prediction, but compresses the sequence of intermediate belief updates that led to it. We find that the *dynamics* of the reverse process provide a complementary signal: variables whose predicted relationships change repeatedly across timesteps tend to participate in overlapping or competing repair configurations. As the diffusion converges, the trajectory preserves information about structural ambiguity encountered along the way. We exploit this signal for neighborhood selection. Empirically, trajectory sensitivity correlates with structural coupling rather than training-specific cost correlations, and exhibits improved robustness across distributions compared to final predictions alone.

**Trajectory Sensitivity Score.** For each set $i$, we define a trajectory sensitivity score that measures the cumulative variation of predicted affinities across diffusion timesteps:

$$\mathcal{S}_i = \frac{1}{m} \sum_{j=1}^{m} \sum_{t=1}^{T-1} \big( \hat{p}_{ij}^{(t+1)} - \hat{p}_{ij}^{(t)} \big)^2, \qquad (9)$$

where $\hat{p}_{ij}^{(t)} \equiv (\hat{X}_{0|t})_{ij}$ denotes the predicted probability that sets $i$ and $j$ belong to the same cluster at timestep $t$, corresponding to the $(i, j)$-th entry of the model's predicted affinity matrix $\hat{X}_{0|t}$. This quantity is a lightweight proxy for uncertainty along the denoising trajectory: it captures how frequently the model revises its structural beliefs before convergence, rather than estimating uncertainty in the final prediction. In practice, this path-variation statistic is computationally inexpensive, stable for Bernoulli predictions, and sufficient for ranking variables by their involvement in unresolved structural ambiguity.

**Sensitivity-Weighted Affinity.** We combine converged predictions with trajectory sensitivity to form the final affinity matrix. Let $\tilde{\mathcal{S}}_i \in [0, 1]$ denote min–max normalized sensitivity. The pairwise affinity is defined as

$$A_{ij}^{\text{sim}} = \hat{p}_{ij}^{(1)} \cdot \big( 1 + \tau \cdot \tilde{\mathcal{S}}_i \tilde{\mathcal{S}}_j \big). \qquad (10)$$

where $\tau$ is temperature parameter. This factorized modulation upweights connections between variables that both exhibit elevated trajectory sensitivity, while leaving low-sensitivity affinities unchanged. The multiplicative form acts as a coordination gate—high instability in a single variable is insufficient on its own—encouraging joint repair only when ambiguity is shared. This design introduces no additional learnable parameters and empirically improves neighborhood stability when extrapolating from bounded training subproblems to larger instances with different cost structures. Appendix C provides an analysis of why trajectory instability correlates with structural ambiguity, and a controlled synthetic verification with planted regimes (essential/exchangeable/coupled/trivial), showing clear separation and regime discriminability (Figs. 3–5).

## 3.5. Partitioning and Sequential Repair

We partition the graph induced by $A^{\text{sim}}$ into $K$ disjoint neighborhoods $\{\mathcal{N}_1, \ldots, \mathcal{N}_K\}$ using METIS (Karypis & Kumar, 1997), which minimizes edge cuts to keep structurally coupled variables together.

We process neighborhoods sequentially, allowing each repair to build on prior improvements. For each neighborhood $\mathcal{N}_k$, we first *destroy* the incumbent solution by deselecting all active sets $u \in \mathcal{N}_k$. We then *repair* the induced subproblem by re-optimizing over $\mathcal{N}_k$ using a steepest-descent local search procedure (Caserta, 2007), while holding all variables $u \notin \mathcal{N}_k$ fixed. The solution is updated only if repair strictly improves the objective, ensuring monotonic cost reduction. This completes one pass of trajectory-guided LNS; we repeat this operation within a given time budget.

# 4. Experiments

Our experimental evaluation investigates neural methods for large-scale SCP in regimes where exact mixed-integer programming (MIP) solvers are computationally impractical and heuristic guidance is required. We focus on two complementary objectives: (i) understanding which modeling paradigms and mechanisms are effective for structured neural guidance in SCP, and (ii) assessing whether these insights transfer to large instances where optimal solutions are unavailable.

To this end, we divide our evaluation into two parts. We first study medium-scale instances ($10^3$–$10^4$ sets) under controlled conditions to isolate the impact of modeling choices and robustness properties (Q1–Q3). We then evaluate zero-shot extrapolation to large instances (over $10^5$ sets) and compare against state-of-the-art MIP and heuristic solvers under strict time budgets, reflecting the regimes that motivate neural approaches in practice (Q4).

- **Q1 (Paradigm Choice):** Do generative, iterative refinement methods based on Large Neighborhood Search outperform purely constructive neural solvers and discriminative, point-estimate neighborhood selection strategies under controlled conditions?

- **Q2 (Robustness to Objective Shift):** Are the learned structural priors robust to systematic shifts in the cost distribution, when trained on a unicost regime and evaluated on others?

- **Q3 (Mechanistic Contribution):** Does exploiting the denoising trajectory of a diffusion model provide more

informative structural priors for neighborhood selection than relying on final-step predictions alone?

- **Q4 (Scalability):** Does the proposed framework scale to large Set Cover instances ($N \geq 10^5$), where exact MIP solvers become impractical?

### 4.1. Evaluation Suite

We use the Rail and Airline Crew Scheduling benchmarks from OR-Lib (Beasley, 1990). To investigate the impact of cost distribution on performance, we evaluate Rail samples under four cost regimes: (i) **Unicost** ($c_j = 1$), (ii) **Uniform** ($c_j \sim \mathcal{U}[1, 100]$), (iii) **Correlated** ($c_j \propto \deg(j)$), and (iv) **Inverse** ($c_j \propto 1/\deg(j)$). Methodological analysis (Q1–Q3) uses $N = 1,000$ instances, while scalability experiments (Q4) evaluate on instances $N \geq 10^5$. We report the standard **Primal Gap (%)** to the optimal or best known solution.

### 4.2. Baselines

We compare against three paradigms: **Constructive (Neural)** solvers, including DIFUSCO (Sun & Yang, 2023) and DIFFUCO (Sanokowski et al., 2024), which construct solutions in a single pass, and a GNN-based SCP baseline (Shafi et al., 2023); **LNS (Heuristic)** baselines (Random, ALNS (Lan et al., 2007)) using stochastic destroy rules; and **LNS (Neural)** baselines (GNN-LNS , RL-LNS (Wu et al., 2021), Diffusco-LNS (Feng et al., 2024)) that employ discriminative models or learned policies for neighborhood selection. All neural models use a 6-layer GAT (Veličković et al., 2017) backbone. All LNS variants share the same local solver and iteration budget.

## 5. Results and Discussion

### 5.1. Comparison of Different Paradigms (Q1)

Table 1 highlights three consistent trends across benchmarks: limitations of one-shot constructive solvers, the strong baseline performance of simple stochastic repair, and the benefits of generative neighborhood modeling.

**Constructive vs. Repair-based Neural Solvers** Purely constructive methods (e.g., DIFUSCO, GNN) consistently underperform, with primal gaps 2–3× higher than even the simplest repair heuristics. This behavior is consistent with the known difficulty of one-shot construction for Set Cover, where early decisions induce long-range constraint interactions that are costly to revise. In contrast, repair-based methods initialize from a feasible greedy solution, substantially reducing the effective search space and allowing neural guidance to focus on local improvement rather than feasibility.

**The effectivness of Random-LNS** As observed in Table 1, `Random-LNS` yields a competitive gap, outperforming several neural policies. This result reinforces prior observations that, for highly constrained problems, the destroy–repair paradigm itself provides a strong inductive bias, even in the absence of learned guidance (Sonnerat et al., 2021).

Our results confirm that the iterative cycle of destruction and re-optimization is inherently suited to the structure of SCP. However, the failure of traditional neural policies (*e.g.*, RL-LNS at $3.10\%$) to beat this random baseline suggests a "learned bias" problem. While Song et al. (2020) showed that LNS policies can be learned for general Integer Programs, our results indicate that in highly constrained, varied-cost SCP instances, these policies often converge to myopic behaviors that lack the diversity required to escape local optima found by random sampling.

**Limitations of Deterministic Neural Guidance.** As shown in Table 1, deterministic neural LNS policies often do not outperform the Random-LNS baseline. A consistent limitation of these approaches is that neighborhood selection is treated as a static point-estimate task: models aim to identify a fixed subset of variables to modify, rather than capturing coordinated uncertainty over multiple repair options. In the highly coupled landscape of SCP, this focus can restrict exploration and lead to premature convergence. Both RL–based policies (RL-LNS) and supervised GNN-based methods (GNN-LNS, CL-LNS) exhibit this behavior, which is reflected in their higher primal gaps.

**Generative Trajectories as a Repair Signal.** While Diffusco-LNS introduces generative sampling, it treats the diffusion model largely as a black box by relying only on final denoised outputs. In contrast, GLNS explicitly exploits prediction dynamics along the denoising trajectory to guide neighborhood selection. This trajectory signal enables GLNS to consistently achieve lower primal gaps than other neural LNS variants as shown in Table 1.

### 5.2. Robustness to Cost Distribution Shift (Q2)

We evaluate zero-shot robustness to cost variation by training all neural models exclusively on *Unicost* instances and evaluating them on three unseen cost regimes. Table 2 reports performance separately for each regime. Across these settings, discriminative baselines exhibit reduced robustness, with performance varying substantially as the cost structure departs from the training distribution.

In contrast, GLNS maintains consistently low primal gaps across all cost regimes, including the challenging *Inverse* setting. This regime is a known stress test for cost-driven heuristics, as cost–degree correlations are reversed relative to the training data. The stability of GLNS across regimes

*Table 1.* Main results on RAIL and SCP datasets. **Gap (%)** is the relative optimality gap. To assess algorithmic sample efficiency, all LNS variants are evaluated using a fixed budget of iterations. **Time** is wall-clock time in seconds. Methods are grouped by paradigm: **Constructive** methods build solutions from scratch; **LNS** methods iteratively refine an initial feasible solution.

| METHOD | RAIL UNICOSTS | | RAIL VARIED COSTS | | AIRLINE CREW SCHEDULING | |
|---|---|---|---|---|---|---|
| | PRIMAL GAP (%) | TIME(S) | PRIMAL GAP (%) | TIME(S) | PRIMAL GAP (%) | TIME(S) |
| *Repair Baselines* | | | | | | |
| RANDOM-LNS | 4.12 | 0.5 | 2.98 | 0.5 | 10.76 | 0.5 |
| ADAPTIVE-LNS | 4.52 | 0.5 | 2.88 | 0.5 | 10.76 | 0.5 |
| *Constructive (Neural)* | | | | | | |
| DIFUSCO | 6.42 | 1.93 | 7.15 | 1.31 | 18.34 | 0.87 |
| DIFFUCO | 6.71 | 1.18 | 7.89 | 1.25 | 19.62 | 0.82 |
| FAST T2T | 5.87 | 1.1 | 5.2 | 1.4 | 11.13 | 0.9 |
| GNN | 5.95 | 0.09 | 6.72 | 0.94 | 16.87 | 0.61 |
| *LNS (Neural)* | | | | | | |
| RL-LNS | 4.5 | 0.7 | 3.1 | 0.81 | 12.56 | 0.9 |
| GNN-LNS | 3.83 | 0.81 | 2.81 | 0.8 | 11.56 | 0.9 |
| CL-LNS | 3.83 | 0.65 | 2.92 | 0.62 | 11.84 | 0.61 |
| DIFFUSCO-LNS | 3.65 | 1.05 | 2.88 | 1.11 | 9.96 | 0.9 |
| GLNS (OURS) | 3.28 | 1.08 | 2.55 | 0.94 | 9.1 | 1.02 |

suggests that its trajectory-guided neighborhood selection captures structural properties of the incidence graph that are less sensitive to the specific cost distribution, whereas point-estimate predictors tend to rely on cost-feature correlations tied to the training regime.

### 5.3. The Role of Trajectory Dynamics in Neighborhood Selection (Q3)

To isolate the contribution of trajectory information, we compare GLNS against three alternative signaling mechanisms while keeping the rest of the LNS pipeline fixed. We consider: (i) STATIC-GNN, which uses cosine similarity between learned node embeddings; (ii) SPECTRAL, which derives neighborhoods from eigenvectors of the set-intersection graph Laplacian; and (iii) DIFFUSION-FINAL, which relies only on the final denoised output of the diffusion model and discards intermediate trajectory information.

As shown in Table 2, all ablation variants consistently underperform the full GLNS model across benchmarks. In particular, DIFFUSION-FINAL—which retains generative sampling but summarizes the denoising process by a single final prediction—yields higher primal gaps than GLNS on datasets such as *Airline*. This indicates that access to intermediate denoising dynamics provides additional structural signal beyond what is captured by the converged prediction alone.

While Table 2 shows that GLNS performs well across benchmarks and cost regimes, we further isolate the effect of trajectory-guided neighborhood selection. For this, Appendix F reports additional ablations in Table 11. There, we compare the trajectory signal against alternatives such as the

final diffusion step, final map entropy, ensemble variance, Monte Carlo dropout, and random size-matched clustering, while keeping diffusion steps, wall-clock budget, similarity construction, and clustering fixed. The results show that the trajectory-based signal is the main source of the gains, rather than the surrounding decomposition or solver pipeline. Overall, these results suggest that prediction dynamics along the denoising trajectory encode information about coordinated variable interactions that is not fully captured by final diffusion outputs in our setting, and that exploiting this signal leads to more effective neighborhood selection.

We further corroborate the stability of our method through ablation and sensitivity analyses in Figure 7. Since GLNS is a multi-stage pipeline, we test its robustness across key design choices, including the diffusion horizon T, METIS partitioning variants, sub-problem solvers, and decomposition granularity, controlled by the cluster count K and maximum cluster size. The results show that GLNS remains robust across these settings: the advantage of the trajectory signal persists without instance-specific tuning.

### 5.4. Scaling (Q4)

To evaluate scalability beyond the training regime, we assess GLNS on large-scale SCP instances with up to $N = 10^5$ sets. All models are trained on instances with sizes $N \in [10^3, 10^4]$ and deployed in a zero-shot manner, without scale-specific retraining or tuning.

At this scale, neural methods that rely on global graph processing or single-pass construction are limited by memory constraints or inference cost. GLNS avoids these bottlenecks by operating on bounded halo subproblems and apply-

*Table 2.* Zero-shot cost transfer performance across diverse cost distributions. We report the Primal Gap (%) relative to the Known Optimal Solution for various SCP variants. Notably, all neural models are trained exclusively on the Unicost distribution and evaluated in a zero-shot setting on others to test distributional robustness. To assess algorithmic sample efficiency, all LNS variants are evaluated using a fixed budget of iterations. GLNS consistently outperforms existing neural LNS baselines and specialized diffusion solvers like Diffusco-LNS. The significant margin on the Airline (OR-Lib) benchmark demonstrates that GLNS captures fundamental structural dependencies that transfer to standard library instances without fine-tuning. GT-LNS is a supervised baseline with access to ground-truth labels indicating which variables differ between the heuristic and optimal solutions.

| Method | Unicost (Train) | Uniform | Correlated | Inverse | Airline (OR-Lib) |
| --- | --- | --- | --- | --- | --- |
| GT-LNS | 3.33±0.01 | 1.47±0.04 | 2.99±0.02 | 2.08±0.06 | 7.16±0.04 |
| *Repair Baselines* | | | | | |
| Random-LNS | 4.12±0.08 | 2.14±0.07 | 3.50±0.05 | 2.85±0.07 | 9.32±0.1 |
| ALNS | 4.52±0.08 | 2.32±0.04 | 3.79±0.02 | 3.13±0.04 | 9.50±0.06 |
| *Neural LNS* | | | | | |
| GNN-LNS | 3.82±0.06 | 2.98±0.07 | 3.74±0.06 | 2.98±0.06 | 12.56±0.66 |
| CL-LNS | 3.81±0.06 | 2.98±0.07 | 3.74±0.06 | 2.98±0.06 | 12.56±0.66 |
| Diffusco-LNS | 3.65±0.03 | 1.89±0.04 | 3.16±0.02 | 2.65±0.09 | 9.72±0.05 |
| *GLNS Ablations* | | | | | |
| Static-GNN | 4.41±0.09 | 2.03±0.07 | 3.57±0.09 | 2.87±2.58 | 8.90±0.03 |
| Spectral | 4.44±0.11 | 2.16±0.15 | 3.67±0.12 | 2.93±0.08 | 10.32±0.07 |
| GLNS-Final | 3.45±0.06 | 1.79±0.04 | 2.96±0.09 | 2.35±0.02 | 9.52±0.06 |
| **GLNS (Ours)** | **3.28**±0.08 | 1.68±0.03 | **2.86**±0.09 | **2.21**±0.04 | **8.36**±0.02 |

*Table 3.* Primal Integral (PI) comparison on large-scale Set Cover instances ($N \geq 10^5$) over a strict 60-second wall-clock budget. PI measures cumulative solution quality over time (lower is better). We include Gurobi with and without presolve to isolate its effect. Our model achieves the lowest PI, reflecting superior anytime performance.

| SOLVER | PI ↓ | RATIO |
| --- | --- | --- |
| **GLNS** | **760.59** | **1.00** |
| GUROBI (PRESOLVE) | 995.57 | 1.31 |
| GUROBI | 1032.05 | 1.36 |
| GREEDY+ALNS | 1084.00 | 1.43 |
| CP-SAT | 2505.00 | 3.29 |
| RANDOM STEEPEST | 4916.64 | 6.46 |
| SCIP | 144041.53 | 189.38 |

ing iterative, trajectory-guided repair. Within a fixed time budget, the method performs multiple refinement passes, each targeting structurally salient regions identified by the diffusion trajectory. Appendix E provides a per-component runtime breakdown for GLNS (Fig. 6), showing that GNN + diffusion inference contribute a constant/diminishing fraction at larger scales, while halo sampling (and clustering) dominates runtime.

**Anytime Performance.** Table 3 reports the Primal Integral (Berthold, 2013) over a 60-second horizon, which aggregates solution quality across all time points. GLNS achieves the lowest PI (760.59), outperforming even Gurobi with presolve enabled (995.57, 1.31×). The gap is more pronounced against CP-SAT and SCIP, which incur substantial overhead before producing competitive feasible solutions.

Notably, Gurobi's presolve provides only marginal improvement (1.31× vs. 1.36×).

**Scalable Transfer.** As problem size increases, the trajectory-based signal remains stable and GLNS maintains a consistent solution quality. In contrast, classical heuristics such as Greedy ALNS plateau early (1.43×), while exact solvers require extended runtime to close the gap.

Solvers like CP-SAT can achieve superior final gaps given sufficient runtime (> 60s) the Primal Integral confirms that GLNS provides more reliable solutions under tight time constraints. This makes GLNS well suited as a front-end or initialization component for hybrid optimization pipelines, where rapid computation of high-quality feasible solutions is critical.

# 6. Conclusion

We presented GLNS, a framework that leverages discrete diffusion trajectories to guide neighborhood selection for large-scale Set Cover. By extracting sensitivity signals from the denoising dynamics—rather than relying solely on final predictions—GLNS identifies structurally coherent repair regions that transfer across cost distributions and scale to large instances without retraining.

GLNS consistently produces high-quality solutions early in the optimization process, achieving strong anytime performance while remaining competitive at longer time horizons. These results indicate that generative trajectory information can serve as an effective structural prior for neighborhood selection, addressing the locality and overfitting limitations

of point-estimate neural guidance. Appendix G evaluates the GLNS pipeline on additional combinatorial problems showing consistent improvements over baselines in a controlled synthetic setting. Several directions remain for future work, including adaptive halo sizing, tighter integration with branch-and-bound solvers, and a deeper theoretical characterization of when trajectory-based signals outperform static embeddings.

## Acknowledgments

We thank the members of the OR-Team at Google Research for their valuable insights, with particular gratitude to Ondrej Sykora and Ferdinand Genans for their helpful discussions throughout this project. We also acknowledge the Center for Cognition and Computation at Goethe University for their support of Achref Jaziri.

## Impact Statement

This paper presents work whose goal is to advance the field of Machine Learning and Combinatorial Optimization. There are many potential societal consequences of our work, none which we feel must be specifically highlighted here.

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

## Supplementary Materials Overview

This appendix provides additional discussion, analysis, and experimental results that complement the main paper. In particular, it (i) clarifies design choices and assumptions underlying GLNS, (ii) provides further intuition for the proposed trajectory-based neighborhood selection mechanism, and (iii) reports extended experimental results, ablations, and runtime analyses that could not be included in the main text due to space constraints.

To facilitate navigation, Table 4 summarizes the contents of each appendix section and highlights where specific questions or concerns are addressed.

| Section | Contents and Purpose |
|---|---|
| Appendix A | Extended discussion of design choices and assumptions, including computational limitations, supervision requirements, solver selection, and applicability beyond Set Cover. |
| Appendix B | Extended related work on neural combinatorial optimization, diffusion models, and solver-integrated learning methods. |
| Appendix C | Empirical analysis and qualitative intuition for trajectory sensitivity, illustrating how diffusion dynamics capture structural ambiguity during neighborhood selection. |
| Appendix D | Model architecture, training details, and implementation specifics required for reproducibility. |
| Appendix E | Runtime breakdown and scaling analysis, including per-component timing and amortization behavior at large problem sizes. |
| Appendix F | Additional experiments and ablations, including comparisons against alternative neighborhood selection signals, hyperparameter sensitivity, and extended problem domains. |

*Table 4.* Overview of appendix contents and where specific questions are addressed.

## A. Extended Discussion

The results in Section 5 demonstrate that trajectory-guided neighborhood selection consistently improves solution quality, robustness to cost shifts, and anytime performance in large-scale Set Cover instances. In this section, we step back from empirical comparisons to discuss the broader implications of these findings. Specifically, we clarify why diffusion models are a natural fit for neighborhood selection, examine the computational trade-offs introduced by iterative generative inference, and delineate the regimes in which GLNS is practically beneficial. We also discuss training requirements, generalization beyond Set Cover, and the rationale behind our evaluation choices. Our goal is not to extend the experimental claims, but to situate GLNS within the design space of neural–symbolic optimization methods and to make explicit the assumptions and limitations that shape its applicability.

### A.1. Why use diffusion for neighborhood selection?

Diffusion is well suited to neighborhood selection because it is a *generative* model whose iterative denoising process exposes prediction instability across timesteps. Variables whose assignments fluctuate during denoising tend to correspond to structurally coupled or ambiguous decisions, making them high-value candidates for joint repair. While diffusion offers a stable and practical framework for this purpose, other uncertainty estimation mechanisms within generative models may provide promising directions for future work.

### A.2. What are the computational limitations of diffusion-based neighborhood selection?

Diffusion-based methods incur additional computational cost due to their iterative inference procedure. In isolation, this cost can be mitigated through standard techniques such as reducing the number of diffusion steps, employing more expressive denoisers, or adopting improved noise schedules and sampling schemes.

In the regime studied in this work, however, diffusion inference does not constitute the primary computational bottleneck. As shown in Figure 5, both GNN inference and diffusion-based refinement contribute a constant or diminishing fraction of the per-iteration runtime as problem size increases. Instead, the dominant cost at scale arises from classical components, in particular the halo sampling procedure used to construct localized subproblems, as well as the subsequent clustering step. These operations scale linearly with instance size by design and are independent of the diffusion formulation.

Consequently, the scalability of GLNS at large problem sizes is not limited by diffusion inference itself, but by controllable preprocessing and solver-side operations. For clarity and modularity, we therefore adopt a standard discrete diffusion

formulation in this work. Exploring more efficient diffusion variants and tighter integration with downstream clustering procedures is a promising direction for future work, but is orthogonal to the core contribution of leveraging diffusion trajectories as a signal for neighborhood selection.

### A.3. Does GLNS require optimal solutions for training?

Partially. The first-stage GNN, which identifies high-recall candidate variables for repair, is trained using supervision derived from optimal or near-optimal solutions obtained via an exact solver. Importantly, this supervision is collected exclusively on small instances, where computing high-quality SCIP solutions is computationally inexpensive (typically on the order of 1–2 seconds per instance) and does not constitute a practical bottleneck for data generation. The central premise of GLNS is precisely to leverage such supervision at small scales in order to generalize to substantially larger problem instances, where exact optimization becomes infeasible.

The diffusion model itself does not rely on optimal labels. It is pretrained to produce structurally coherent clusterings and subsequently refined using cost feedback from the symbolic repair solver during inference. As a result, GLNS avoids dependence on optimal supervision at the target scales of interest, while still benefiting from inexpensive solver guidance during training on small instances.

### A.4. In which settings is GLNS practically beneficial?

GLNS targets large-scale optimization, with instances containing $10^5$–$10^7$ variables and time budgets of 10–60 seconds. In this regime, exact solvers often struggle to produce strong feasible solutions quickly, while simple heuristics plateau early. GLNS is also effective as a *warm-start generator*: solutions obtained within 5–10 seconds can substantially improve downstream exact solver performance. For small instances or relaxed time budgets, commercial solvers remain preferable.

### A.5. Does GLNS generalize beyond Set Cover?

While the main body of this work focuses on Set Cover, the core mechanism underlying GLNS—using diffusion trajectory dynamics to guide iterative large neighborhood search—is not specific to a single problem formulation. In particular, GLNS does not rely on handcrafted heuristics tied to Set Cover, but instead operates on a generic bipartite graph representation and a problem-agnostic generative refinement process.

To assess whether this procedure extends beyond Set Cover, we evaluate GLNS on additional NP-hard combinatorial optimization problems, including Maximum Coverage, Set Packing, and Vertex Cover. In these experiments, GLNS is trained from scratch for each problem using the same architecture and training protocol, with only problem-specific feasibility checks and greedy repair operators adapted. As shown in Appendix G, GLNS consistently outperforms greedy and random LNS baselines across all evaluated problem types, and benefits from the full diffusion trajectory rather than relying solely on the final denoised solution.

These results suggest that the effectiveness of GLNS stems from its ability to identify and refine structurally related subsets of variables through iterative generative updates, a principle that naturally extends to a broad class of constraint-based optimization problems. We therefore focus on Set Cover in the main text to enable a rigorous and controlled analysis of the proposed mechanisms, while the additional experiments demonstrate that the approach applies more broadly beyond this specific domain.

### A.6. Why were these particular MIP solvers chosen for evaluation?

We evaluate against Gurobi, CP-SAT, and SCIP to cover the three dominant paradigms for exact Set Cover solving. Gurobi represents best-in-class commercial MIP performance, CP-SAT emphasizes constraint programming and strong propagation, and SCIP provides a widely used open-source MIP baseline. This selection enables a fair comparison across solver philosophies while maintaining reproducibility.

# B. Extended Related Work

### B.1. Scale Generalization in Neural Combinatorial Optimization

A persistent challenge in neural combinatorial optimization is the limited ability of learned models to generalize beyond their training distribution in terms of problem size. Fu et al. (Fu et al., 2021) show that models trained on TSP-50 can be extended to larger instances through a divide-and-conquer strategy that recursively partitions problems into smaller subproblems. Joshi et al. (Joshi et al., 2020) demonstrate that standard Graph Neural Networks fail to generalize to larger graphs due to fixed receptive fields, and propose architectural modifications such as hierarchical pooling to mitigate this limitation. Relatedly, Kool et al. (Kool et al., 2022) analyze attention-based solvers and find that generalization depends sensitively on normalization schemes and positional encodings, with models often learning size-specific biases that do not transfer.

A common thread in successful scaling strategies for routing problems is the exploitation of geometric locality: optimal solutions tend to connect spatially proximate nodes, enabling effective decomposition into quasi-independent subproblems. The Set Cover Problem fundamentally lacks such geometric structure. Its bipartite incidence graph induces global coupling, where a single variable assignment can affect constraints across distant regions of the problem. As a result, SCP presents a stringent testbed for scale generalization: effective approaches cannot rely on spatial priors and must instead identify structural decomposition strategies that remain valid across diverse instance topologies.

Our halo-based decomposition addresses this challenge by extracting localized subproblems using problem-specific heuristics—such as replaceability scores and bipartite neighborhood expansion—while bounding subproblem size. This ensures that the neural model operates on a consistent problem structure regardless of the global instance scale.

### A.2 Neural Large Neighborhood Search: Methods and Limitations

Large Neighborhood Search (LNS) (Shaw, 1998) iteratively improves solutions by destroying subsets of variables and repairing the resulting partial solutions using a sub-solver. Its effectiveness depends critically on neighborhood selection: neighborhoods must be sufficiently large to escape local optima, yet structured enough to allow efficient repair. Adaptive LNS (Ropke & Pisinger, 2006) extends this framework by maintaining portfolios of destroy operators selected via bandit-based strategies, highlighting the importance of diversity in neighborhood selection.

Neural approaches to LNS differ primarily in how neighborhoods are selected. Classification-based methods (Song et al., 2020; Huang et al., 2023) train models to predict which variables to destroy, typically using supervised learning on solution trajectories. These approaches collapse inherently uncertain repair decisions into deterministic point estimates, discarding information about confidence and variable interdependence. Reinforcement learning approaches (Wu et al., 2021) formulate neighborhood selection as a Markov Decision Process, but often suffer from high variance and sample inefficiency. Autoregressive methods (Hottung & Tierney, 2019) sequentially construct neighborhoods by conditioning each selection on previous choices, enabling flexible neighborhood sizes but introducing ordering biases that can compound early errors.

Recent work has explored diffusion models for combinatorial optimization. DIFUSCO (Sun & Yang, 2023) demonstrates that discrete diffusion can generate high-quality solutions for problems such as TSP and Maximum Independent Set by framing solution construction as an iterative denoising process. DIFUSCO-LNS (Feng et al., 2024) extends this idea to LNS by training diffusion models in a supervised setting to imitate branching decisions from solvers, then sampling neighborhoods from the final denoised output. While these methods show that generative models can capture complex solution structure, they treat the diffusion process itself as a black box, extracting only the final prediction and discarding the rich sequence of intermediate states generated during denoising.

### B.2. Uncertainty Quantification and Importance Estimation

Uncertainty and importance estimation have long played a central role in machine learning. The Fisher Information Matrix provides an information-geometric measure of parameter importance, quantifying how model predictions change under perturbations (Amari, 1998). Kirkpatrick et al. (Kirkpatrick et al., 2017) leverage this concept in Elastic Weight Consolidation (EWC), using diagonal Fisher estimates to identify neural network parameters critical for previously learned tasks. By penalizing changes to high-Fisher parameters, EWC mitigates catastrophic forgetting in continual learning. While this framework focuses on parameter importance, it provides a useful conceptual analogue for identifying important problem variables in optimization settings.

Recent work has also examined uncertainty within diffusion models themselves. Graikos et al. (Graikos et al., 2022) show

that intermediate predictions during the reverse diffusion process can serve as uncertainty estimates for inverse problems in computer vision. By measuring variance across diffusion timesteps, they identify regions where the model's predictions are unstable and correlate this instability with reconstruction error. Their results establish that diffusion trajectories contain information beyond the final sample. However, their application focuses on continuous-domain reconstruction quality rather than discrete optimization guidance.

We build on this insight by adapting trajectory-based uncertainty to combinatorial optimization. In our setting, prediction instability across diffusion timesteps serves as a learned heuristic for identifying structurally coupled variables that require coordinated repair. Unlike ensemble-based uncertainty estimation, trajectory variance captures convergence dynamics within a single model and requires no repeated inference. This makes it particularly attractive for large-scale optimization, where computational budgets are tightly constrained.

### B.3. Sensitivity Analysis in Mathematical Optimization

Classical optimization theory provides formal notions of variable and constraint importance through sensitivity analysis (Bertsimas & Tsitsiklis, 1997; Wolsey, 2020). In linear programming, dual variables quantify how the optimal objective changes under constraint perturbations, while reduced costs indicate the impact of introducing non-basic variables. These measures underpin variable fixing and pruning strategies in Mixed-Integer Programming solvers. However, they require access to optimal or near-optimal solutions with dual certificates—information typically unavailable during heuristic search on large instances.

This gap motivates the development of learned proxies for structural importance. Prior neural combinatorial optimization work has explored attention weights (Kool et al., 2018), embedding similarities, and gradient-based measures as relevance signals. Our trajectory sensitivity score contributes to this line of work by deriving importance from generative model dynamics rather than discriminative predictions. We do not claim to recover classical sensitivity measures, which have precise definitions tied to optimality conditions. Instead, we propose a learned heuristic that empirically correlates with structural importance and remains informative across problem scales where exact sensitivity analysis is infeasible.

### B.4. Set Cover Problem

The Set Cover Problem is NP-hard. The classical greedy algorithm of Chvátal (Chvatal, 1979) achieves an $O(\log n)$ approximation guarantee and performs well on many practical instances, though it can be arbitrarily suboptimal in worst-case constructions. Alternative approaches include LP relaxation and rounding, as well as local search methods that trade approximation guarantees for empirical performance.

Neural approaches to Set Cover remain limited. Shafi et al. (Shafi et al., 2023) propose a supervised Graph Neural Network that predicts set inclusion probabilities, but focuses on small instances. Recently, DIFUSCO-LNS (Feng et al., 2024) incorporated SC into their evaluation with instances ranging from 4,000 to 8,000 sets. While their method leverages a diffusion model to capture the multimodal nature of LNS destroy policies , it relies on imitation learning, requiring expert demonstrations from Local Branching to train the policy network. More broadly, SC has seen limited adoption in neural combinatorial optimization benchmarks: small instances are efficiently solved by exact methods, while large instances pose severe challenges for both classical and neural approaches.

## C. Empirical Analysis and Intuition for Trajectory Sensitivity

This appendix provides a theoretical interpretation of the trajectory sensitivity signal used by GLNS. Its purpose is not to establish optimality guarantees, but to explain *why* prediction instability along the reverse diffusion process serves as a meaningful proxy for structural ambiguity in combinatorial optimization.

We frame diffusion-based inference over Set Cover solutions as approximate posterior sampling from an energy-based distribution and analyze the behavior of posterior mean predictions across diffusion timesteps. Under this view, we show that trajectory-level instability arises naturally in two settings: (i) variables with high posterior entropy, corresponding to exchangeable or locally neutral decisions, and (ii) variables that are strongly coupled to such ambiguous choices. Conversely, variables that are structurally essential or dominated exhibit stable trajectories.

We further connect these probabilistic observations to the discrete geometry of the Set Cover landscape by relating trajectory sensitivity to notions of tightness, replaceability, and coupled constraint structure. This yields a taxonomy of variable roles

that explains when large-neighborhood repairs are likely to be effective and motivates the design of trajectory-weighted affinity matrices used in the main algorithm. Throughout, the arguments are heuristic and interpretive. Their role is to provide mechanistic grounding for the empirical results in the main paper and to clarify the link between diffusion dynamics and effective neighborhood selection in large-scale discrete optimization. Crucially, this appendix analyzes a simplified vector-based diffusion model (predicting variable inclusion) rather than the higher-order matrix model (predicting pairwise affinities) used in the main GLNS algorithm.

### C.1. Setup and Definitions

We begin by establishing the probabilistic framing that connects diffusion-based inference to combinatorial optimization.

**Definition 1** (Target Distribution over Solutions). Let $\mathcal{X}^* \subseteq \{0,1\}^m$ be the set of feasible solutions to a Set Cover instance with $m$ sets. We define the target distribution over solutions as:

$$p^*(\mathbf{X}) \propto \exp(-\beta \cdot J(\mathbf{X})) \cdot \mathbf{1}[\mathbf{X} \text{ is feasible}] \tag{11}$$

where $J(\mathbf{X}) = \boldsymbol{c}^\top \mathbf{X}$ is the objective function, $\boldsymbol{c} \in \mathbb{R}_+^m$ is the cost vector, and $\beta > 0$ controls concentration around optimal solutions. This defines an *energy landscape* where the "energy" of a configuration is its objective value $J(\mathbf{X})$.

This Boltzmann formulation is standard in the combinatorial optimization literature and forms the basis of simulated annealing (Kirkpatrick et al., 1983). As $\beta \to \infty$, the distribution concentrates on optimal solutions. The energy landscape perspective allows us to reason about optimization geometry: low-energy regions correspond to high-quality solutions, and the curvature of the energy surface reflects the sensitivity of solution quality to local perturbations.

**Assumption 1** (Distributional Approximation). The learned denoiser $f_\theta$, trained on high-quality solutions obtained from exact or near-optimal solvers, approximates the conditional expectation under the target distribution $p^*$. Specifically, for the posterior mean prediction at timestep $t$:

$$\hat{\mathbf{X}}_{0|t} \approx \mathbb{E}_{p^*}[\mathbf{X}_0 | \mathbf{X}_t, \mathcal{G}] \tag{12}$$

This assumption is standard in the neural combinatorial optimization literature, where models are trained via supervised learning on solver-generated solutions (Gasse et al., 2019) or reinforcement learning with cost-based rewards (Kool et al., 2018).

**Definition 2** (Posterior Mean Trajectory). Let $\hat{\mathbf{X}}_{0|t} \in [0,1]^m$ denote the model's soft prediction of the clean solution at diffusion timestep $t$, conditioned on the noisy state $\mathbf{X}_t$ and the problem graph $\mathcal{G}$:

$$\hat{\mathbf{X}}_{0|t} = \mathbb{E}_\theta[\mathbf{X}_0 | \mathbf{X}_t, \mathcal{G}] \tag{13}$$

Each component $\hat{X}_{0|t}^{(i)} \in [0,1]$ represents the model's estimated probability that set $i$ is selected in the optimal solution, given the current noisy observation. The *denoising trajectory* is the sequence $\{\hat{\mathbf{X}}_{0|t}\}_{t=T}^1$ obtained by running the reverse diffusion process.

We index the reverse diffusion process such that $t = T$ corresponds to the initial noisy state (high noise, low signal) and $t = 1$ corresponds to the final denoised output (low noise, high signal). The trajectory proceeds $T \to T - 1 \to \cdots \to 1$. This convention aligns with our implementation where the denoising loop iterates from $T$ down to 1.

It is essential that $\hat{\mathbf{X}}_{0|t}$ represents soft (continuous) predictions rather than hard (binary) assignments. If the model produced only binary outputs, the difference $\hat{X}_{0|t+1}^{(i)} - \hat{X}_{0|t}^{(i)}$ would be restricted to $\{-1, 0, 1\}$, reducing the sensitivity metric to a simple count of state flips. The continuous representation captures the nuanced "confidence wobble" of the model—gradual shifts in predicted probabilities that reveal structural uncertainty even when the final prediction is decisive.

**Definition 3** (Trajectory Sensitivity). For variable $i \in \{1, \ldots, m\}$, the *trajectory sensitivity* is defined as:

$$S_i = \frac{1}{T-1} \sum_{t=1}^{T-1} \left( \hat{X}_{0|t+1}^{(i)} - \hat{X}_{0|t}^{(i)} \right)^2 \tag{14}$$

This measures the cumulative instability of predictions for variable $i$ across the denoising process, capturing how much the model's belief about variable $i$ fluctuates as noise is progressively removed.

## C.2. Main Conceptual Result

This appendix provides an interpretive perspective on trajectory sensitivity, aimed at offering intuition rather than formal guarantees. This analysis illustrates how denoising dynamics can be viewed through an energy-based lens that aligns with the observed empirical behavior

## C.3. Trajectory Sensitivity as an Uncertainty Proxy

Let $H_i = H(X_{0,i} \mid \mathcal{G})$ denote the marginal entropy of variable $i$ under the target distribution $p^*$, and let $I_{ij} = I(X_{0,i}; X_{0,j} \mid \mathcal{G})$ denote the mutual information between variables $i$ and $j$. We posit that high trajectory sensitivity $S_i$ is indicative of two complementary structural properties:

1. **Intrinsic Ambiguity:** High $H_i$ (the variable participates in multiple competing optima).

2. **Strong Coupling:** High $I_{ij}$ to an ambiguous variable $j$ (the variable is structurally dependent on an unstable decision).

*Heuristic Justification.* The following argument provides intuition linking diffusion dynamics to equilibrium properties of $p^*$. We rely on the assumption that the learned denoiser approximates the posterior expectation, $\hat{\mathbf{X}}_{0|t} \approx \mathbb{E}_\theta[\mathbf{X}_0 \mid \mathbf{X}_t]$.

**Regime 1: Low Entropy (Stability).** Consider a variable $i$ where $p^*(X_{0,i} \mid \mathcal{G})$ is highly concentrated (e.g., $X_{0,i} = 1$ in almost all valid solutions). In this case, the posterior remains dominated by the concentrated prior even at early diffusion timesteps (large $t$). The conditional expectation $\mathbb{E}_\theta[X_{0,i} \mid \mathbf{X}_t]$ stays close to the prior mean across the trajectory because $\mathbf{X}_t$ rarely contains evidence strong enough to contradict the overwhelming prior. Consequently, the trajectory $\{\hat{X}_{0|t}^{(i)}\}_t$ is stable, resulting in small trajectory sensitivity $S_i$.

**Regime 2: High Entropy (Instability via Phase Transition).** Consider a variable $i$ with high marginal entropy (e.g., $p^*(X_{0,i} = 1) \approx 0.5$). The denoising process undergoes a phase transition:

- **High Noise ($t \approx T$):** The observation $\mathbf{X}_t$ is uninformative. The optimal estimator outputs the marginal mean $\approx 0.5$.

- **Low Noise ($t \to 0$):** The noise is small enough that the model collapses to a specific mode (0 or 1) determined by stochastic aspects of the reverse process.

Crucially, in the *intermediate* regime, the estimator is highly sensitive to the specific realization of the noise $\epsilon_t$. Small fluctuations in $\mathbf{X}_t$ can tip the posterior belief between competing modes. This "wobble" as the model resolves ambiguity creates large cumulative temporal variation, resulting in high $S_i$.

**Regime 3: Induced Instability via Coupling.** Finally, consider a variable $i$ with moderate entropy but strong dependence on an unstable variable $j$ (high $I_{ij}$). The denoiser learns the conditional structure of the problem. If the model's belief about variable $j$, denoted $\hat{X}_{0|t}^{(j)}$, oscillates due to the mechanism described in Regime 2, this oscillation propagates to variable $i$ through the learned correlations. If $\hat{X}_{0|t}^{(i)}$ is locally approximated as $f(\hat{X}_{0|t}^{(j)})$ for some learned dependency $f$, then:

$$\text{Var}_t(\hat{X}_{0|t}^{(i)}) \approx \left( \frac{\partial f}{\partial \hat{X}_{0|t}^{(j)}} \right)^2 \text{Var}_t(\hat{X}_{0|t}^{(j)}) \tag{15}$$

where $\text{Var}_t$ denotes empirical variance across diffusion timesteps. Thus, high sensitivity $S_j$ induces high sensitivity $S_i$ proportional to the strength of their coupling.

**Summary.** $S_i$ acts as a mechanistic detector for these phenomena: it aggregates the magnitude of the estimator's response to progressively reduced noise, which peaks precisely in regions of high posterior uncertainty or strong coupling.     $\square$

## C.4. Geometric Interpretation of Trajectory Sensitivity

We now provide a geometric interpretation that connects probabilistic notions of stability and uncertainty to the combinatorial structure of the Set Cover Problem. We argue that the trajectory sensitivity $S_i$ acts as a probe that is sensitive to specific features of the underlying discrete optimization landscape induced by $J(\mathbf{X})$.

To articulate this connection, we classify variables based on their role in maintaining feasibility and their potential for substitution relative to a local optimum $\mathbf{X}^*$.

**Definition 4** (Solution-Relative Tightness)**.** Let $\mathbf{X}^*$ be a locally optimal solution. The *tightness* of an active set $s$ (where $X_s^* = 1$) is the number of elements covered *uniquely* by $s$:

$$T(s) = \left| \left\{ e \in \mathcal{U} : A_{se} = 1 \text{ and } \sum_{j \neq s} A_{je} X_j^* = 0 \right\} \right| \tag{16}$$

Note that $T(s)$ is defined relative to the specific configuration $\mathbf{X}^*$ and characterizes the immediate structural infeasibility induced by removing set $s$.

**Definition 5** (Replaceability)**.** The *replaceability* of an active set $s$ is the count of inactive sets that can individually substitute for $s$ without increasing cost:

$$R(s) = \left| \left\{ j : X_j^* = 0, \text{ cov}(j) \supseteq \text{unique-cov}(s), \ c_j \leq c_s \right\} \right| \tag{17}$$

where $\text{cov}(j) = \{e : A_{je} = 1\}$ denotes the coverage of set $j$, and unique-cov$(s)$ denotes the set of elements uniquely covered by $s$ in $\mathbf{X}^*$. This definition captures direct single-set swaps; it does not account for complex multi-set replacements.

**Remark 1** (Local Geometry)**.** The condition $R(s) > 0$ with $c_j = c_s$ implies that the solution lies on a locally flat region of the optimization landscape along the coordinate $s$, where the objective value is invariant to specific discrete changes.

We interpret trajectory sensitivity $S_i$ as providing a soft signal correlated with the balance between a variable's tightness $T(s)$ and its replaceability $R(s)$. This interaction defines three distinct structural regimes:

**1. The Essential Regime (High Stability).** Consider a variable where $T(s)$ is high and $R(s) = 0$.

- **Structure:** The variable is structurally load-bearing. Removing it renders the solution infeasible, and no valid single-set replacement exists.

- **Dynamics:** The learned denoising dynamics strongly favor the feasible configuration ($X_s = 1$). The model assigns consistently low probability to the alternative state across the generation process, resulting in a stable trajectory.

- **Result:** $S_i$ is low.

**2. The Exchangeable Regime (High Ambiguity).** Consider a variable where $T(s)$ is low and $R(s)$ is high (many equal-cost swaps).

- **Structure:** The solution exhibits local neutrality. The specific choice of including set $s$ versus a neighbor $j$ is arbitrary with respect to the objective function.

- **Dynamics:** The model lacks a strong signal preferring $s$ over its neighbors. Consequently, the prediction exhibits noise-driven drift, influenced significantly by stochastic injections at each timestep.

- **Result:** $S_i$ is high.

**3. The Coupled Regime (Combinatorial Symmetry).** Consider a cluster of variables $\mathcal{C}$ that are mutually exclusive but collectively necessary (e.g., satisfying a constraint that requires exactly one of $k$ sets).

- **Structure:** The landscape contains multiple valid configurations separated by combinatorial barriers (rather than continuous energy barriers). The variables are structurally interdependent: satisfying one constraint forces a specific value on another, creating a chain of dependencies.

- **Dynamics:** As noise levels decrease, the model must break symmetry to select one valid configuration. In the intermediate inference stages, the model often oscillates between competing valid assignments because the variables are statistically coupled.

- **Result:** $S_i$ is high for all variables in the cluster due to this coupled instability.

LNS is most effective when it repairs the mutable parts of a solution while preserving the core structure. By selecting variables with high $S_i$, our method preferentially targets the Exchangeable and Coupled regimes—regions of ambiguity or symmetry breaking—where local re-optimization is most likely to yield improvement or escape local optima.

## C.5. Synthesis: A Trajectory-Based Structural Taxonomy

We now synthesize the heuristic arguments from Section C.3 into a unified interpretive taxonomy. This taxonomy maps the observable signal—trajectory sensitivity $S_s$—to the underlying combinatorial role of variables, guiding the design of the LNS repair operator.

Let $T(s)$ and $R(s)$ be the tightness and replaceability relative to a candidate solution $\mathbf{X}^*$, and let $S_s$ be the observed trajectory sensitivity. We categorize variables into four regimes based on the interaction between structural features and model dynamics:

(i) **The Essential Regime (High Stability):**

- **Profile:** High $T(s)$, Low $R(s) \implies$ Low $S_s$.
- **Characterization:** The variable is structurally critical with no viable alternatives. The posterior concentrates strongly on inclusion ($X_s = 1$), reflecting a locally stable configuration.
- **LNS Implication:** *Preserve.* Excluding $s$ from repair neighborhoods is desirable, as modifying it would likely break feasibility.

(ii) **The Exchangeable Regime (High Ambiguity):**

- **Profile:** Low $T(s)$, High $R(s) \implies$ High $S_s$.
- **Characterization:** The variable is one of several equivalent options. The model-induced posterior is diffuse, reflecting a locally flat landscape where the prediction may drift between swap candidates.
- **LNS Implication:** *Explore.* Including $s$ in a repair neighborhood allows the solver to pivot between equivalent configurations, potentially escaping local optima.

(iii) **The Coupled Regime (Coupled Clusters):**

- **Profile:** High $T(s)$ and strong conditional dependence, typically associated with high $S_s$.
- **Characterization:** The variable belongs to a cluster $\mathcal{C}$ where decisions are interdependent (symmetry breaking). The trajectory oscillates due to coupled instability.
- **LNS Implication:** *Co-optimize.* The entire cluster $\mathcal{C}$ should be included in the same repair neighborhood. Treating these variables independently often fails to resolve the deadlock.

(iv) **The Deterministic Regime (Low Sensitivity):**

- **Profile:** Low $T(s) \implies$ Low $S_s$.
- **Characterization:** The variable corresponds to a structurally trivial high-confidence decision that is either always excluded (dominated sets) or trivially required (unconstrained mandatory sets). The model assigns consistently extreme probabilities (near 0 or 1) across timesteps.
- **LNS Implication:** *Ignore.* These variables lie far from the optimization frontier and require no intervention.

The taxonomy relies on the correspondence between landscape geometry and diffusion dynamics. **Regime (i)** corresponds to the stable signal dominated by the prior. **Regime (ii)** corresponds to the noise-driven drift observed in high-entropy locally neutral regions. **Regime (iii)** captures the coupled oscillations induced by strong dependencies. **Regime (iv)** represents the case where the signal-to-noise ratio is high because the model assigns consistently extreme probabilities, resulting in a stable, low-sensitivity trajectory.

The classification above motivates our design of the trajectory-weighted affinity matrix:

$$A_{ij}^{\mathrm{sim}} = \hat{p}_{ij}^{(1)} \cdot (1 + \tilde{S}_i \tilde{S}_j) \tag{18}$$

where $\tilde{S}$ is the normalized sensitivity. The term $\tilde{S}_i \tilde{S}_j$ acts as multiplicative gating, upweighting connections only when *both* variables exhibit high sensitivity. High sensitivity correlates with coupling. Therefore, spectral clustering on $A^{\mathrm{sim}}$ tends, in practice, to place members of coupled clusters $\mathcal{C}$ into the same partition, enabling the LNS solver to perform the coordinated moves necessary to improve the solution.

## C.6. Controlled Empirical Verification

To empirically validate the structural taxonomy, we design a controlled synthetic experiment of SCP instances. The goal is to isolate the mechanism behind trajectory sensitivity—independent of instance scale, solver heuristics, or dataset artifacts—and test whether diffusion trajectory instability aligns with known structural roles by construction.

**Synthetic benchmark**  We generate Set Cover instances over $n = 100$ elements with a fixed number of sets $m = 40$. Each instance is partitioned into four disjoint groups, where the structural role of every set is known a priori:

1. **Essential** ($s \in \mathcal{S}_{\text{ess}}$): sets that uniquely cover at least one element and must be selected in any feasible solution.

2. **Exchangeable** ($s \in \mathcal{S}_{\text{exch}}$): pairs of sets with identical incidence patterns and equal costs, such that either choice is optimal, inducing true local degeneracy.

3. **Coupled** ($s \in \mathcal{S}_{\text{coup}}$): a dependency motif with $k$ sets spanning $k-1$ elements, where each set omits exactly one element and any $(k-1)$ suffice for coverage, creating interdependent ambiguity.

4. **Trivial** ($s \in \mathcal{S}_{\text{triv}}$): dominated, high-cost sets that overlap only with elements already covered by essential sets and are never selected.

In our generator, we instantiate $|\mathcal{S}_{\text{ess}}| = 12$, $|\mathcal{S}_{\text{exch}}| = 12$ (6 exchangeable pairs), $|\mathcal{S}_{\text{coup}}| = 8$, and $|\mathcal{S}_{\text{triv}}| = 8$.

**Model and evaluation protocol.**  We train a lightweight GNN denoiser under the discrete Bernoulli diffusion formulation, using $T = 50$ diffusion steps. The model predicts clean selections $x_0$ from noisy states $x_t$ via binary cross-entropy training. Evaluation is performed on 256 unseen test instances.

**Trajectory sensitivity.**  For each test instance, we sample $K = 10$ stochastic reverse trajectories (distinct noise seeds) and record the predicted inclusion probabilities $\hat{p}^{(t)} \in [0, 1]^m$ at every diffusion step. We quantify per-set trajectory instability using

$$S_i \;=\; \sum_{t=1}^{T-1} \left( \hat{p}_i^{(t)} - \hat{p}_i^{(t-1)} \right)^2, \tag{19}$$

averaged over trajectories. Intuitively, $S_i$ is small when the reverse process is stable and decisive, and large when predictions fluctuate across timesteps.

**Results: regime separation.**  The resulting distributions of $S_i$ are shown in Figure 1. The observed ordering matches theory:

$$\text{Trivial} \;<\; \text{Essential} \;<\; \text{Coupled} \;<\; \text{Exchangeable},$$

with minimal overlap between regimes. Exchangeable variables concentrate at high sensitivity, while Essential and Trivial variables remain tightly clustered near zero.

**Statistical discrimination.**  Using $S_i$ as a scalar score, we evaluate regime discrimination via ROC analysis. As shown in Figure 2, $S_i$ achieves strong separability across all theoretically relevant comparisons (e.g., Essential vs. Exchangeable, Trivial vs. Coupled), with AUROC values well above chance. This confirms that trajectory sensitivity is not only statistically significant but also practically discriminative.

**Trajectory-level interpretation.**  Representative denoising trajectories are visualized in Figure 3. Essential variables become decisive early and remain stable, Exchangeable variables exhibit pronounced flips and late commitment, Coupled variables show intermediate instability due to correlated alternatives, and Trivial variables remain suppressed throughout. Notably, some Exchangeable variables end with confident final predictions despite exhibiting high trajectory instability, illustrating why final-step uncertainty alone is insufficient.

This controlled experiment provides direct empirical validation of the core mechanism underlying GLNS: *trajectory sensitivity reliably identifies structurally ambiguous and replaceable decisions*. The diffusion reverse process is most unstable on exchangeable variables—precisely those for which large-neighborhood perturbations are most meaningful—while remaining stable on essential and dominated variables. These signals are not recoverable from final-step predictions alone, motivating the use of trajectory-guided neighborhood generation.

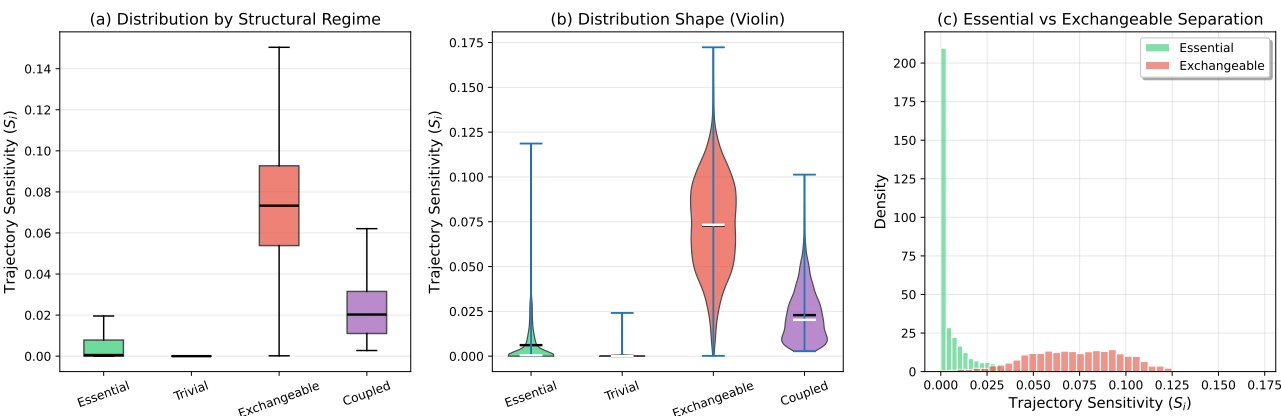

*Figure 1.* **Trajectory sensitivity across different regimes. (a)** Box plots and **(b)** violin plots of $S_i$ show clear separation between regimes. **(c)** Overlaid histograms for Essential vs. Exchangeable highlight strong discriminability, indicating that $S_i$ captures replaceability that is not visible from final-step confidence alone.

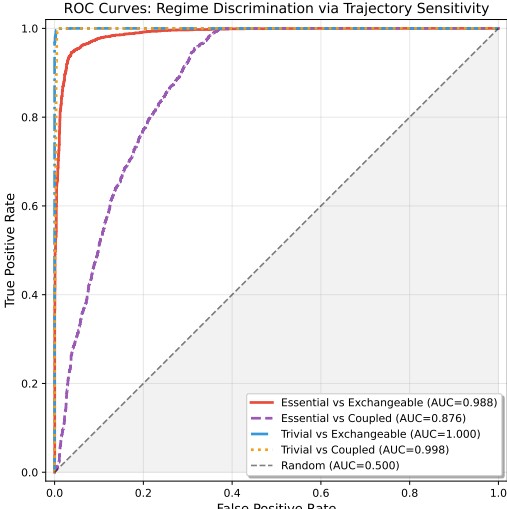

*Figure 2.* **ROC curves for regime discrimination using $S_i$.** Trajectory sensitivity provides strong separation across multiple regime pairs, supporting its role as a structural ambiguity signal.

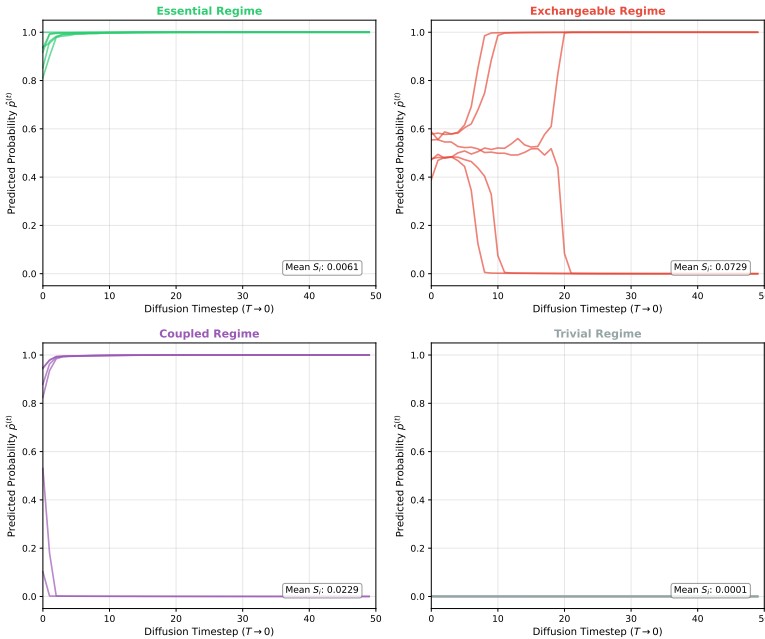

*Figure 3.* **Representative denoising trajectories by regime.** Predicted inclusion probabilities $\hat{p}_i^{(t)}$ over diffusion timesteps ($T \to 0$) for example variables in each planted regime. Exchangeable variables exhibit strong trajectory instability, while Essential and Trivial variables remain stable.

# D. Implementation Details

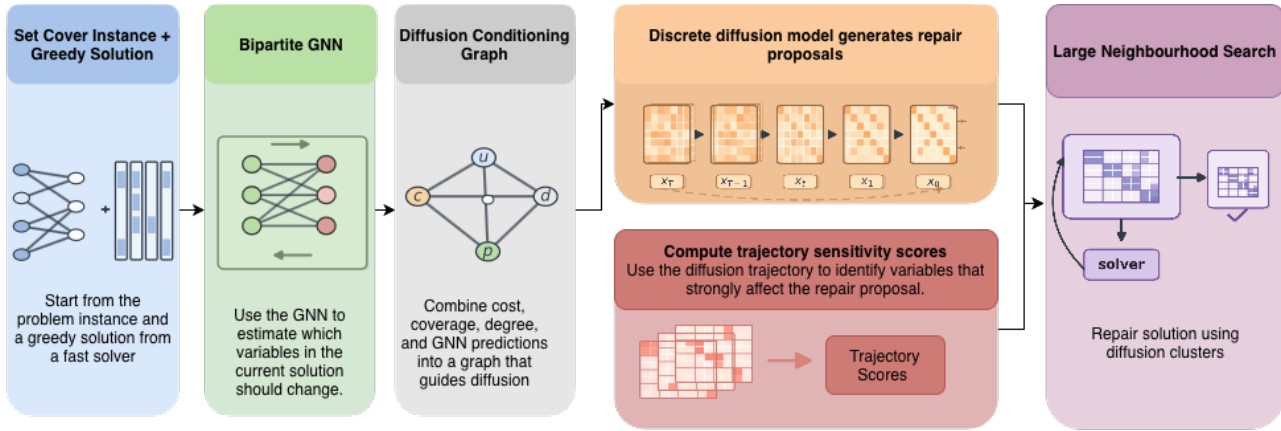

*Figure 4.* Overview of the GLNS repair pipeline. Given a set cover instance and a fast greedy solution, a bipartite GNN predicts variables likely to change. These predictions are combined with cost, coverage, and degree features to condition a discrete diffusion model, which generates coordinated repair proposals and trajectory sensitivity scores. The resulting clusters define destroy-and-repair neighborhoods for a large-neighborhood search solver, which updates the incumbent solution when an improved repair is found.

## D.1. Dataset Generation

Training instances are derived from the RAIL benchmark suite in the OR-Library (Beasley, 1990). Each source instance is downsampled to produce problem instances with at most $N = 1{,}000$ sets and $M = 500$ elements. We generate 50 random samples per source file by varying the random seed for downsampling, yielding approximately 10,000 training instances per cost structure. The dataset is partitioned into training, validation, and test sets using an 80/10/10 split ratio with fixed random seeds to ensure reproducibility.

To evaluate structural generalization, we generate four cost variants for each problem instance. Let $\deg(v_j)$ denote the

number of elements covered by set $j$.

- **Unicost:** All sets have unit cost: $c_j = 1$ for all $j \in \mathcal{S}$.

- **Uniform:** Costs are sampled independently from a continuous uniform distribution:

$$c_j \sim \mathcal{U}(1, 100). \tag{20}$$

- **Correlated:** Costs are positively correlated with set coverage, penalizing high-degree sets:

$$c_j = 10 + 80 \cdot \frac{\deg(v_j)}{\max_k \deg(v_k)} + \eta, \quad \eta \sim \mathcal{U}(1, 20). \tag{21}$$

- **Inverse:** Costs are inversely proportional to coverage, encouraging selection of high-degree sets:

$$c_j = \max\left(1, \ 90 - 70 \cdot \frac{\deg(v_j)}{\max_k \deg(v_k)} + \eta\right), \quad \eta \sim \mathcal{U}(1, 20). \tag{22}$$

Optimal or near-optimal solutions are computed using the **SCIP** solver (Bolusani et al., 2024). We enforce a 60-second time limit per instance; solutions that remain feasible within this window are retained for training.

Initial solutions for each instance are generated using a two-phase heuristic with Element Degree Greedy Solver for greedy construction, followed by Steepest Search local improvement with a maximum of 500 iterations. Both heuristics are implemented in Perron & Furnon (2025).

For the in-distribution experiments, models are trained separately on each cost structure. For the zero-shot transfer experiments, all neural models are trained exclusively on the Unicost distribution and evaluated without fine-tuning on the remaining cost regimes.

### D.2. Bipartite GNN for State-Change Prediction

This section details the architecture and training procedure for the first-stage GNN, which is trained separately to predict which sets should change between the heuristic and optimal solutions.

#### D.2.1. INPUT FEATURES

The GNN operates on a bipartite graph $G = (\mathcal{S} \cup \mathcal{E}, A)$ where $\mathcal{S}$ denotes set nodes and $\mathcal{E}$ denotes element nodes. All features are normalized using Z-score standardization.

**Set Node Features.** Each set $j \in \mathcal{S}$ is represented by a 3-dimensional feature vector:

- **Normalized Cost:** $\tilde{c}_j = (c_j - \mu_c)/\sigma_c$

- **Normalized Unit Cost:** $(c_j / \deg(v_j) - \mu_u)/\sigma_u$, capturing cost efficiency relative to coverage

- **In-Solution Indicator:** Binary flag $x_j \in \{0, 1\}$ indicating whether the set is selected in the heuristic solution

**Element Node Features.** Each element $e \in \mathcal{E}$ is represented by a 3-dimensional feature vector:

- **Normalized Degree:** Number of sets covering the element (constraint tightness)

- **Normalized Min Neighbor Cost:** Cost of the cheapest set covering this element

- **Normalized Mean Neighbor Cost:** Average cost of all sets covering this element

Learnable type embeddings of dimension $d = 256$ are added to distinguish set nodes from element nodes.

*Table 5.* Bipartite GNN architecture hyperparameters.

| Hyperparameter | Value |
| --- | --- |
| Hidden dimension $d$ | 256 |
| Message-passing layers | 6 |
| Attention heads | 8 |
| Dropout rate | 0.1 |
| Edge embedding dimension | 64 |
| Skip connections | Last 3 layers |

### D.2.2. ARCHITECTURE

We implement a GATv2-style (Brody et al., 2021) graph attention network with the specifications summarized in Table 5.

We augment the bipartite incidence graph with several additional structural connections. First, we include reverse element-to-set edges to enable bidirectional message passing between sets and elements. Second, we add self-loops to all nodes to preserve local information across layers. Finally, we introduce a global virtual node that is connected to all nodes in the graph, allowing constant-time ($O(1)$) global information exchange and facilitating the propagation of instance-level context. Learnable edge type embeddings are used to distinguish the five edge types: forward, reverse, self-loop, virtual-to-node, and node-to-virtual.

Each layer applies pre-LayerNorm, multi-head attention with learned edge features, and a feed-forward network (FFN) with GELU activation and expansion factor 4. Residual connections are applied around both the attention and FFN blocks. Multi-scale skip connections concatenate outputs from the final three layers before projecting to the readout head.

A three-layer MLP with LayerNorm and GELU activations ($256 \rightarrow 128 \rightarrow 1$) produces per-node logits. During training, a mask ensures that only set nodes contribute to the loss; element nodes are ignored.

### D.2.3. TRAINING OBJECTIVE

The model minimizes a composite loss combining Focal Loss and Tversky Loss with equal weighting ($\lambda = 0.5$):

$$\mathcal{L} = \lambda \, \mathcal{L}_{\text{Tversky}} + (1 - \lambda) \, \mathcal{L}_{\text{Focal}}. \tag{23}$$

**Focal Loss.** Focal Loss (Lin et al., 2017) down-weights well-classified examples to focus learning on hard cases:

$$\mathcal{L}_{\text{Focal}} = -\frac{1}{|\mathcal{M}|} \sum_{j \in \mathcal{M}} \alpha_t (1 - p_t)^\gamma \log(p_t), \tag{24}$$

where $p_t$ is the predicted probability for the true class, $\gamma = 2$ is the focusing parameter, and $\alpha_t$ is a dynamic class-balancing weight computed as the ratio of negative to positive samples, clipped to $[1, 100]$.

**Tversky Loss.** Tversky Loss (Salehi et al., 2017) generalizes Dice loss with asymmetric weighting to prioritize recall:

$$\mathcal{L}_{\text{Tversky}} = 1 - \frac{\text{TP} + \epsilon}{\text{TP} + \alpha \cdot \text{FN} + \beta \cdot \text{FP} + \epsilon}, \tag{25}$$

where $\alpha = 0.3$ and $\beta = 0.7$. This configuration penalizes false negatives (missed repair candidates) more heavily than false positives, ensuring high recall. The downstream symbolic solver can efficiently discard false positives during the repair phase.

### D.2.4. OPTIMIZATION

Training hyperparameters are summarized in Table 6.

We maintain an exponential moving average (EMA) of model parameters with decay rate 0.999. The EMA parameters are used for validation evaluation and final model selection. Training is monitored using the mean repaired cost on the validation set. If no improvement is observed for 5 consecutive evaluations, early stopping is triggered.

*Table 6.* Bipartite GNN training hyperparameters.

| Hyperparameter | Value |
|---|---|
| Optimizer | AdamW |
| Weight decay | $10^{-5}$ |
| Learning rate schedule | Warmup + Cosine decay |
|    Initial learning rate | $10^{-6}$ |
|    Peak learning rate | $10^{-3}$ |
|    Warmup steps | 1,000 |
|    Total training steps | 100,000 |
| Batch size | 32 |
| Gradient clipping (max norm) | 1.0 |
| EMA decay | 0.999 |
| Early stopping patience | 5 evaluations |
| Train / Validation split | 90% / 10% |
| Random seed | 42 |

## D.3. Diffusion Model for Structural Inference

This section details the discrete diffusion model that infers latent repair clusters by generating pairwise similarity matrices over candidate sets.

### D.3.1. TARGET REPRESENTATION

The diffusion model learns to predict a symmetric binary clustering matrix $X_0 \in \{0, 1\}^{N \times N}$, where $X_{0,ij} = 1$ indicates that sets $i$ and $j$ belong to the same repair cluster.

Given the change mask from the first-stage GNN, we partition sets into clusters as follows:

- **Agreement cluster:** All sets where the heuristic agrees with the predicted optimal (change mask $= 0$) are assigned to a single background cluster (Cluster 0).

- **Disagreement clusters:** Sets predicted to change (change mask $= 1$) are clustered using the Louvain community detection algorithm (Blondel et al., 2008) on the set-set overlap graph, with resolution parameter $\gamma = 0.8$.

The target matrix $X_0$ is constructed by setting $X_{0,ij} = 1$ if sets $i$ and $j$ belong to the same cluster (including self-loops on the diagonal), and symmetrizing via $X_0 \leftarrow \max(X_0, X_0^\top)$.

### D.3.2. NOISE SCHEDULE

We employ a discrete Bernoulli diffusion process. The forward process progressively corrupts the clean clustering matrix $X_0$ by randomly flipping entries according to a noise schedule.

At timestep $t$, each entry of $X_t$ is independently corrupted with probability $(1 - \bar{\alpha}_t)$, where corrupted entries are replaced with uniform random bits:

$$q(X_t|X_0) = \text{Bernoulli}\left(X_t; X_0\bar{\alpha}_t + \tfrac{1}{2}(1 - \bar{\alpha}_t)\right), \tag{26}$$

where $\bar{\alpha}_t = \prod_{i=1}^{t} \alpha_i$ and $\alpha_t = 1 - \beta_t$.

We use a linear schedule with $T = 100$ timesteps:

$$\beta_t = \beta_1 + \frac{t - 1}{T - 1}(\beta_T - \beta_1), \quad \beta_1 = 10^{-4}, \quad \beta_T = 0.05. \tag{27}$$

The reverse process reconstructs $X_0$ by iteratively sampling $X_{t-1}$ from $X_t$. At each timestep, the denoiser predicts $\hat{X}_0$, and

the posterior probability is computed as:

$$P(X_{t-1} = 1|X_t, \hat{X}_0) = \frac{\sqrt{\bar{\alpha}_{t-1}}\beta_t}{1 - \bar{\alpha}_t}\hat{X}_0 + \frac{\sqrt{\bar{\alpha}_t}(1 - \bar{\alpha}_{t-1})}{1 - \bar{\alpha}_t}X_t, \tag{28}$$

clipped to $[0, 1]$ for numerical stability.

### D.3.3. DENOISER ARCHITECTURE

The denoiser is a Graph Attention Network (Veličković et al., 2017) that predicts $\hat{X}_0$ given the noisy state $X_t$, the conditioning graph $G_d$, and the timestep $t$.

The input graph $G_d = (\mathcal{S}, E)$ is constructed from the set-set overlap structure:

- **Nodes:** Each set $j$ has a 4-dimensional feature vector: normalized cost $\tilde{c}_j$, number of elements covered, element rarity (mean inverse coverage), and the first-stage GNN prediction.

- **Edges:** Weighted by the number of shared elements between sets; only non-zero overlaps are included.

The noisy matrix $X_t$ is concatenated row-wise to the node features, yielding input dimension $(4 + N)$ per node.

Sinusoidal positional embeddings encode the diffusion timestep $t$:

$$\text{PE}(t)_{2i} = \sin(t/10000^{2i/d}), \quad \text{PE}(t)_{2i+1} = \cos(t/10000^{2i/d}), \tag{29}$$

where $d = 128$ is the embedding dimension. The timestep embedding is broadcast and concatenated to all node features.

The denoiser consists of:

- An initial linear projection to dimension $d = 128$

- 3 GAT layers with residual connections and ReLU activations

- Self-loops added to each node for stable message passing

- Dropout with rate 0.1 applied after the initial projection (training only)

The predicted clustering matrix is computed via inner products of the final node embeddings:

$$\hat{X}_0 = \sigma(HH^\top), \tag{30}$$

where $H \in \mathbb{R}^{N \times d}$ are the final node representations and $\sigma$ denotes the sigmoid function applied element-wise.

### D.3.4. TRAINING PROTOCOL

Training proceeds in two phases: supervised pre-training followed by reinforcement learning fine-tuning.

The model is trained to reconstruct the target clustering matrix from noisy inputs using a composite loss:

$$\mathcal{L}_{\text{pretrain}} = \mathcal{L}_{\text{recon}} + \lambda_{\text{sym}}\mathcal{L}_{\text{sym}} + \lambda_{\text{trans}}\mathcal{L}_{\text{trans}}. \tag{31}$$

**Reconstruction Loss.** Masked binary cross-entropy on the upper triangle with dynamic class balancing:

$$\mathcal{L}_{\text{recon}} = \frac{1}{|\mathcal{M}|}\sum_{(i,j)\in\mathcal{M}} w_{ij} \cdot \text{BCE}(\hat{X}_{0,ij}, X_{0,ij}), \tag{32}$$

where $\mathcal{M}$ denotes valid (unpadded) entries in the upper triangle, and $w_{ij} = n_{\text{neg}}/n_{\text{pos}}$ for positive entries and 1 otherwise.

**Symmetry Loss.** Encourages the predicted matrix to be symmetric:

$$\mathcal{L}_{\text{sym}} = \frac{1}{|\mathcal{M}|}\sum_{(i,j)\in\mathcal{M}} \left(\sigma(\hat{X}_{0,ij}) - \sigma(\hat{X}_{0,ji})\right)^2. \tag{33}$$

**Transitivity Loss.** Enforces clustering consistency via hard triplet mining. For each of $K = 256$ sampled intermediate nodes $j$, we identify the hardest incident pairs $i = \arg\max_i P_{ij}$ and $k = \arg\max_k P_{jk}$, then penalize transitivity violations:

$$\mathcal{L}_{\text{trans}} = \frac{1}{K} \sum_j \max(0, P_{ij} + P_{jk} - P_{ik} - 1). \tag{34}$$

After pre-training, the model is fine-tuned using solver feedback to directly optimize repair quality. For each predicted clustering, we run the full LNS repair procedure and measure the cost change:

$$r = -(c_{\text{final}} - c_{\text{initial}}) \times 10, \tag{35}$$

where the scaling factor amplifies the gradient signal.

We maintain an exponential moving average baseline $b$ with decay 0.9:

$$b \leftarrow 0.9 \cdot b + 0.1 \cdot \bar{r}, \tag{36}$$

where $\bar{r}$ is the mean reward over the batch. The advantage is $A = r - b$.

The RL objective modulates the reconstruction loss by the advantage:

$$\mathcal{L}_{\text{RL}} = -\mathbb{E}\left[A \cdot \log P(X_0 | X_t, G_d)\right]. \tag{37}$$

During fine-tuning, the total loss is:

$$\mathcal{L}_{\text{RL-phase}} = \lambda_{\text{sym}}\mathcal{L}_{\text{sym}} + \lambda_{\text{trans}}\mathcal{L}_{\text{trans}} + \lambda_{\text{RL}}\mathcal{L}_{\text{RL}}. \tag{38}$$

### D.3.5. HYPERPARAMETERS

Table 7 summarizes the architecture hyperparameters, and Table 8 summarizes the training configuration.

*Table 7.* Diffusion model architecture hyperparameters.

| Hyperparameter | Value |
|---|---|
| Embedding dimension $d$ | 128 |
| GAT layers | 3 |
| Dropout rate | 0.1 |
| Diffusion timesteps $T$ | 100 |
| Noise schedule | Linear |
| $\beta_1$ | $10^{-4}$ |
| $\beta_T$ | 0.05 |

## D.4. Inference Pipeline

This section details the inference procedure for both in-distribution instances (matching training scale) and large-scale instances requiring hierarchical decomposition.

### D.4.1. TRAJECTORY SENSITIVITY COMPUTATION

During the reverse diffusion process, we record the predicted probabilities $\hat{p}_{ij}^{(t)}$ at each timestep $t \in \{T, T-1, \ldots, 1\}$. The trajectory sensitivity score captures how much each variable's predictions fluctuate during denoising, serving as a proxy for structural uncertainty.

We quantify trajectory sensitivity as the cumulative squared change in predictions across timesteps:

$$\mathcal{V}_{ij} = \sum_{t=1}^{T-1} \left(\hat{p}_{ij}^{(t+1)} - \hat{p}_{ij}^{(t)}\right)^2. \tag{39}$$

*Table 8.* Diffusion model training hyperparameters.

| Hyperparameter | Phase 1 (Pretrain) | Phase 2 (RL) |
|---|---|---|
| Optimizer | AdamW | AdamW |
| Weight decay | $10^{-5}$ | $10^{-5}$ |
| Learning rate | $10^{-4}$ | $10^{-5}$ |
| Batch size | 8 | 8 |
| Epochs | 20 | 25 |
| Steps per epoch | 400 | 400 |
| *Loss weights* | | |
| $\lambda_{\text{sym}}$ | 0.4 | 0.4 |
| $\lambda_{\text{trans}}$ | 0.5 | 0.5 |
| $\lambda_{\text{RL}}$ | — | 0.5 |
| *Other* | | |
| Triplet samples $K$ | 256 | 256 |
| Louvain resolution | 0.8 | 0.8 |
| EMA baseline decay | — | 0.9 |

This measures how much the model's belief about the relationship between sets $i$ and $j$ fluctuates during the denoising trajectory. Variables involved in tightly coupled constraints or multi-modal regions of the solution space typically exhibit higher instability, as the model resolves competing structural pressures.

The node-level sensitivity aggregates pairwise instability:

$$\mathcal{S}_i = \frac{1}{N} \sum_{j=1}^{N} \mathcal{V}_{ij}. \tag{40}$$

Normalizing by $N$ ensures the signal remains scale-invariant across different subproblem sizes.

The final affinity matrix combines the converged predictions with trajectory sensitivity via a rank-1 multiplicative boost:

$$A_{ij}^{\text{sim}} = \hat{p}_{ij}^{(1)} \cdot \left( 1 + \tilde{\mathcal{S}}_i \cdot \tilde{\mathcal{S}}_j \right), \tag{41}$$

where $\tilde{\mathcal{S}}_i \in [0, 1]$ denotes min-max normalized sensitivity. This formulation upweights affinities between pairs of variables that both exhibited high prediction instability, encouraging them to be placed in the same repair neighborhood. The factorized form $\tilde{\mathcal{S}}_i \cdot \tilde{\mathcal{S}}_j$ avoids introducing additional learnable parameters, improving stability when extrapolating to larger instances.

### D.4.2. GRAPH PARTITIONING

The affinity matrix $A^{\text{sim}}$ is used to partition variables into repair neighborhoods.

We construct a weighted graph where edge weights are derived from the similarity matrix:

$$w_{ij} = \lfloor 10000 \cdot A_{ij}^{\text{sim}} \rfloor, \tag{42}$$

with edges below the 20th percentile threshold removed. METIS (Karypis & Kumar, 1997) partitions this graph into $K$ clusters while minimizing edge cut, which groups structurally coupled variables together.

### D.4.3. NEURAL-GUIDED LNS REPAIR

Given the predicted clusters, we perform iterative destroy-and-repair operations.

For each halo, repair proceeds in the following way. For each cluster $C_k$ in sequence:

1. Destroy: Deselect all currently selected sets in $C_k$

2. Repair: Run ElementDegreeSolutionGenerator to restore feasibility

3. Polish: Run SteepestSearch for up to 100 iterations

After each stage and each cluster repair, the solution is only retained if it improves the objective. Otherwise, the previous state is restored.

### D.4.4. HALO-BASED DECOMPOSITION FOR LARGE INSTANCES

For instances exceeding the training scale ($N > 1000$ sets), we employ a hierarchical decomposition strategy that extracts localized subproblems (halos) for neural inference.

We identify promising repair regions by scoring active sets based on their replaceability—the ratio of a set's cost to the average cost of its neighbors in the set-set sharing graph:

$$r(s) = \frac{c_s}{\bar{c}_{\mathcal{N}(s)} + \epsilon}, \quad \bar{c}_{\mathcal{N}(s)} = \frac{\sum_{s' \in \mathcal{N}(s)} c_{s'}}{|\mathcal{N}(s)|}, \tag{43}$$

where $\mathcal{N}(s)$ denotes all sets sharing at least one element with $s$.

Seeds are sampled from active sets proportionally to softmax-transformed replaceability scores:

$$p(s) \propto \exp(r(s)/\tau), \quad s \in \{i : x_i = 1\}, \tag{44}$$

where $\tau = 0.1$ controls the exploration-exploitation tradeoff.

$k$-**Hop Neighborhood Expansion.** Starting from seed sets $\mathcal{S}_0$, the halo is iteratively expanded through the bipartite set-element graph:

$$\mathcal{E}_t = \{e : \exists s \in \mathcal{S}_{t-1}, A_{se} = 1\}, \tag{45}$$
$$\mathcal{S}_t = \{s : \exists e \in \mathcal{E}_t, A_{se} = 1\}. \tag{46}$$

After $k = 2$ iterations, $\mathcal{H} = \bigcup_{t=0}^{k} \mathcal{S}_t$. If $|\mathcal{H}|$ exceeds the target size $N_{\max} = 1000$, non-seed sets are uniformly subsampled.

*Table 9.* Halo sampling hyperparameters.

| Parameter | Value |
|---|---|
| Halos per batch | 8 |
| Target halo size $N_{\max}$ | 1000 |
| Seeds per halo | 5 |
| Expansion hops $k$ | 2 |
| Temperature $\tau$ | 0.1 |
| Minimum neighbors for seed eligibility | 100 |

### D.5. Baseline Configurations and Implementation Details

To ensure a fair and controlled evaluation, we standardize architectural capacity, training protocols, and inference procedures across all neural baselines. Implementations are derived from official repositories where available, with hyperparameters aligned to isolate the impact of each method's core algorithmic contribution rather than incidental implementation differences.

### D.5.1. BASELINE DESCRIPTIONS

**Constructive Neural Methods.** These approaches generate complete solutions in a single forward pass without iterative refinement.

- **GNN** (Shafi et al., 2023): A supervised Graph Neural Network trained to predict set inclusion probabilities. The model operates on the bipartite set–element graph and outputs per-variable logits, which are thresholded and completed via greedy repair to ensure feasibility. Supervision is derived from near-optimal solutions obtained offline using standard heuristics.

- **DIFUSCO** (Sun & Yang, 2023): A graph-based diffusion model that generates binary solution vectors through iterative denoising. The model learns to reverse a discrete corruption process and produces solution candidates that are subsequently repaired to satisfy coverage constraints.

- **DIFFUCO** (Sanokowski et al., 2024): An unsupervised diffusion-based combinatorial optimization method that incorporates constraint penalties directly into the training objective, reducing reliance on optimal solution labels.

**LNS with Heuristic Neighborhood Selection.** These methods iteratively refine solutions using hand-crafted destroy operators.

- **Random-LNS**: Selects variables uniformly at random for destruction. Despite its simplicity, this baseline is known to be competitive in constrained optimization due to the inherent power of repeated destroy–repair cycles (Sonnerat et al., 2021), and serves as a strong reference point.

- **ALNS** (Adaptive Large Neighborhood Search) (Ropke & Pisinger, 2006): Maintains a portfolio of destroy operators (random, worst-cost, and structure-based) and adaptively selects among them based on the performance for previous iterations. Operator weights are updated every 100 iterations with a decay factor of 0.8.

**LNS with Neural Neighborhood Selection.** These methods learn to identify promising subsets of variables for joint repair.

- **GNN-LNS**: Uses a supervised GNN to score variables by their likelihood of differing between the current solution and a near-optimal reference. The top-$k$ scoring variables are destroyed at each iteration. The GNN shares the architecture described in Table 10.

- **CL-LNS** (Huang et al., 2023): Employs contrastive learning to train a neighborhood selector. Positive pairs consist of variables that jointly participate in successful repairs, while negative pairs are sampled from unsuccessful neighborhoods. The learned embeddings guide variable selection via nearest-neighbor clustering.

- **RL-LNS** (Wu et al., 2021): Formulates neighborhood selection as a Markov Decision Process and trains a policy network via REINFORCE. The reward signal is the cost improvement achieved after repair. We use the authors' recommended entropy regularization coefficient of 0.01.

- **Diffusco-LNS** (Feng et al., 2024): Combines diffusion-based generation with LNS by sampling candidate solutions from the denoised output and defining the destroy set as the symmetric difference with the incumbent solution. This baseline inherits architectural components from DIFUSCO to isolate the impact of the search strategy.

### D.5.2. ARCHITECTURAL STANDARDIZATION

To ensure that performance differences reflect algorithmic contributions rather than capacity mismatches, we standardize the following components across all neural methods.

**Graph Neural Network.** All GNN-based methods (GNN, GNN-LNS, CL-LNS, and the encoders used in diffusion-based models) share identical architectural hyperparameters: three message-passing layers, hidden dimension 64, sum aggregation, and dropout rate 0.1. This configuration matches the original DIFUSCO implementation.

**Diffusion Process.** Both DIFUSCO and Diffusco-LNS employ discrete Bernoulli diffusion with $T = 100$ timesteps and a cosine noise schedule.

**Training Protocol.** All neural methods are trained using the AdamW optimizer with learning rate $1 \times 10^{-4}$, weight decay $1 \times 10^{-5}$, and gradient clipping at norm 1.0. Training continues until validation performance plateaus, with an early stopping patience of five epochs.

D.5.3. INFERENCE AND SOLUTION COMPLETION

**Constructive Methods.** For GNN, DIFUSCO, and DIFFUCO, the model outputs a probability vector $\hat{p} \in [0,1]^m$ indicating the likelihood of each set's inclusion. We apply a threshold $\tau = 0.5$ to obtain an initial binary solution. Since this solution may violate coverage constraints, a greedy completion step selects additional sets (in order of cost-efficiency) until feasibility is achieved.

**LNS Methods.** All LNS variants share the same repair subroutine to ensure fair comparison:

1. **Destroy**: Remove selected variables from the incumbent solution.

2. **Repair**: Apply ELEMENTDEGREEGREEDY to restore feasibility.

3. **Polish**: Run STEEPESTDESCENT for up to 100 iterations.

4. **Accept**: Update the incumbent solution only if cost strictly improves.

The destroy set size is fixed at $k = \min(50, 0.1 \cdot |\text{active sets}|)$ for all neural LNS methods. Random-LNS and ALNS use the same budget. This standardization ensures that performance differences stem from neighborhood selection quality rather than destroy magnitude.

None of the evaluated methods, including GLNS, have access to ground-truth or optimal solutions during inference; all decisions are made solely based on the incumbent solution and model predictions.

D.5.4. EVALUATION PROTOCOL

**Iteration Budget.** For controlled experiments (Q1–Q3), all LNS methods execute the same number of destroy–repair iterations, isolating algorithmic sample efficiency from wall-clock considerations.

**Time Budget.** For scaling experiments (Q4), methods operate under a strict 60-second wall-clock limit. Neural methods include inference time in this budget. All methods use identical hardware (single NVIDIA A100 GPU, 32 CPU cores).

**Metrics.** We report Primal Gap (%) relative to the best known solution:

$$\text{Gap} = \frac{c_{\text{method}} - c_{\text{best}}}{c_{\text{best}}} \times 100. \tag{47}$$

For anytime evaluation, we additionally report the Primal Integral (Berthold, 2013), which aggregates solution quality across the optimization horizon.

# E. Runtime Breakdown and Scaling Behavior

To analyze the computational profile of GLNS, we examine how runtime is distributed across its main components as problem size increases (Figure 5). We observe a clear separation between components whose cost scales with instance size and those whose cost remains effectively constant. In particular, both GNN inference and diffusion-based refinement contribute a diminishing fraction of the per-iteration runtime as the number of sets grows, indicating that neural inference costs are amortized and do not dominate large-scale instances. In contrast, the dominant cost at scale arises from halo sampling, which is a deliberately linear-time preprocessing step used to construct localized subproblems. Importantly, solver-side repair (local SCIP optimization) remains a bounded fraction of the total runtime across all scales considered. These results demonstrate that the scalability of GLNS is not limited by neural or diffusion components, but instead by controllable classical operations, enabling the proposed trajectory-based guidance to be applied efficiently at large problem scales.

# F. Ablations and Extended Experiments

## F.1. Is trajectory sensitivity necessary for effective neighborhood selection?

We investigate whether the proposed trajectory-based sensitivity signal is essential for effective neighborhood selection, or whether simpler alternatives based on static uncertainty, epistemic uncertainty, or random grouping could achieve comparable

*Table 10.* Hyperparameter configurations for neural baselines. All methods share GNN architecture and training settings to ensure fair comparison. Method-specific parameters are noted where applicable.

| Category | Parameter | Value | Applies To |
|---|---|---|---|
| **Problem** | Constraint Penalty ($\lambda$) | 100.0 | DIFFUCO |
| | Feasibility Repair | Greedy | All |
| **GNN Architecture** | Hidden Dimension | 64 | All neural |
| | Layers (Depth) | 3 | All neural |
| | Dropout | 0.1 | All neural |
| | Aggregation | Sum | All neural |
| **Diffusion** | Transition Kernel | Bernoulli | DIFUSCO, Diffusco-LNS |
| | Timesteps ($T$) | 100 | DIFUSCO, Diffusco-LNS |
| | Schedule | Cosine / Linear | See text |
| **Training** | Optimizer | AdamW | All neural |
| | Learning Rate | $1 \times 10^{-4}$ | All neural |
| | Weight Decay | $1 \times 10^{-5}$ | All neural |
| | Gradient Clipping | 1.0 | All neural |
| **LNS Repair** | Destroy Size ($k$) | $\min(50, 0.1n)$ | All LNS |
| | Polish Iterations | 100 | All LNS |
| | Accept Criterion | Strict improvement | All LNS |
| **Method-Specific** | Entropy Coefficient | 0.01 | RL-LNS |
| | Contrastive Temperature | 0.07 | CL-LNS |

*Table 11.* Ablation study on Set Cover. We report mean $\pm$ standard deviation of percentage improvement over the greedy baseline, along with average runtime per instance. All methods use identical LNS budgets, neighborhood sizes, and solver configurations; only the neighborhood selection signal is varied. Error statistics reflect variability across problem instances (not across multiple runs with different random seeds).

| Method | % Improvement vs Greedy |
|---|---|
| **GLNS (Trajectory)** | **18.21 $\pm$ 3.56** |
| Final-Step Entropy | 8.29 $\pm$ 3.84 |
| Ensemble Variance ($M$=5) | 6.36 $\pm$ 2.88 |
| MC Dropout ($M$=10) | 5.82 $\pm$ 2.45 |
| Final-Step Affinity | 4.84 $\pm$ 1.98 |
| Random Size-Matched Clusters | 3.19 $\pm$ 2.28 |

performance. To this end, we replace the trajectory signal in GLNS with a series of alternative signals while keeping the remainder of the pipeline fixed, including the GNN-based candidate filter, neighborhood sizes, clustering procedure, solver budgets, and stopping criteria.

We consider three classes of alternatives: (i) *final-step uncertainty measures*, including entropy of the final diffusion marginal (A2.1) and affinity computed from final-step representations (A2.2); (ii) *epistemic uncertainty baselines*, including variance across multiple diffusion runs with different noise seeds (A3.1, $M$=5) and Monte Carlo dropout at inference time (A3.2, $M$=10), where we estimate per-variable uncertainty via variance over stochastic forward passes; and (iii) a *random size-matched clustering* baseline (A1), which controls for the effect of grouping variables without any learned structural signal.

Figure 6 and Table 11 summarize the results on the Set Cover benchmark. While all alternative signals provide modest improvements over random clustering, none approach the performance of trajectory-based GLNS. In particular, trajectory sensitivity achieves a mean improvement of $18.21\%$ over the greedy baseline, more than doubling the best-performing final-step uncertainty method (A2.1), and substantially outperforming both epistemic uncertainty estimates (A3.1–A3.2). The gap is consistent across instances, as reflected by the separation of distributions in Figure 6.

Notably, increasing uncertainty estimation compute does not close this gap. For example, the ensemble baseline (A3.1) incurs substantially higher runtime (Table 11) yet still underperforms trajectory-based selection by a wide margin. Together, these findings suggest that the temporal evolution of diffusion states encodes structural coupling information that is not captured by static final-step uncertainty nor by epistemic variance across stochastic inference runs, supporting the necessity

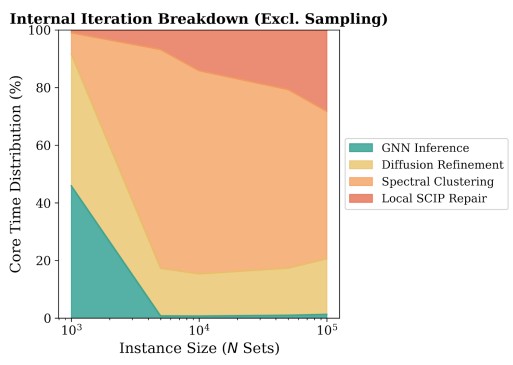

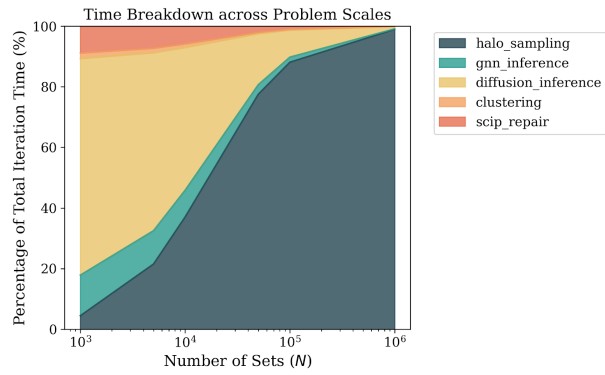

(a) Per-iteration breakdown (excl. sampling)

(a) Per-iteration breakdown (with sampling)

*Figure 5.* Runtime breakdown of GLNS across problem sizes. Neural components (GNN inference and diffusion refinement) incur constant or diminishing relative cost as instance size increases, while overall scaling is dominated by halo sampling, a linear-time classical preprocessing step. Solver-side repair remains a bounded fraction of runtime across all scales.

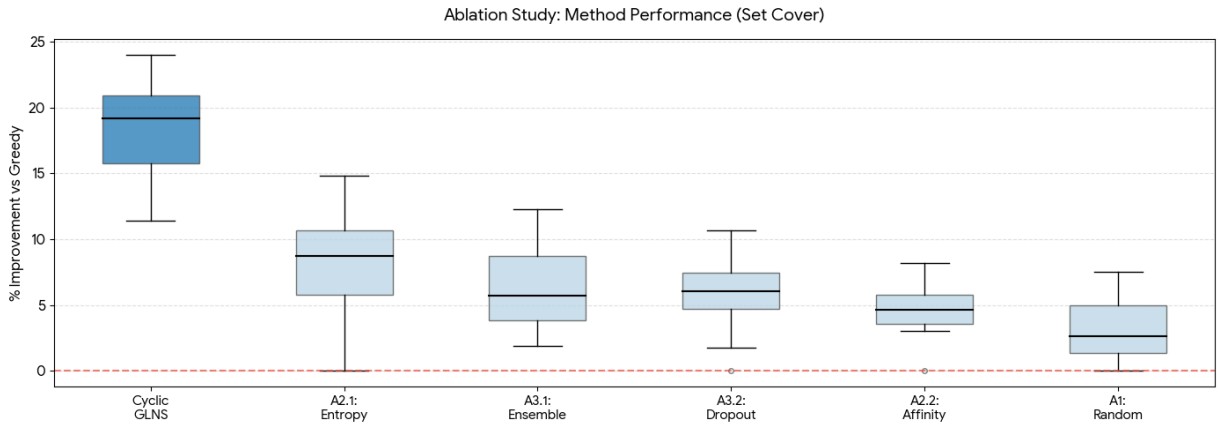

*Figure 6.* Ablation study on Set Cover comparing trajectory-based GLNS against alternative neighborhood selection signals. The boxplots show per-instance percentage improvement over the greedy baseline. Trajectory-based GLNS consistently outperforms final-step uncertainty measures, ensemble-based epistemic uncertainty (including Monte Carlo dropout), and random size-matched clustering. The dashed horizontal line denotes the greedy baseline.

of trajectory-level information for effective neighborhood selection in GLNS.

### F.2. Framework Stability and Design Ablations

We characterize the sensitivity of GLNS to its hyperparameters and architectural choices, summarizing the key findings in Figure 7. These experiments isolate the impact of structural decomposition, diffusion horizons, and the discriminative power of the trajectory-based scores.

**Structural Decomposition and Locality.** The efficacy of iterative repair is fundamentally dependent on the preservation of structural locality during decomposition. As shown in Figure 7b, graph-aware partitioners (METIS, Louvain) significantly outperform geometry-agnostic K-Means by approximately 25 percentage points in gap closure. This confirms that grouping variables by topological coupling—rather than raw feature similarity—is critical for effective symbolic repair.

Performance remains robust across a range of cluster counts ($10 \le K \le 15$, Figure 7d), though overly fine-grained partitioning ($K > 40$) leads to a sharp reduction in gap closure as sub-problems become too small to allow for meaningful combinatorial re-optimization. Similarly, Figure 7e indicates that gap closure plateaus once the maximum cluster size $M_{\max}$ reaches 150. This supports our selection of $M_{\max} = 200$ to balance the repair capacity with the requirement for constant-time complexity per local step.

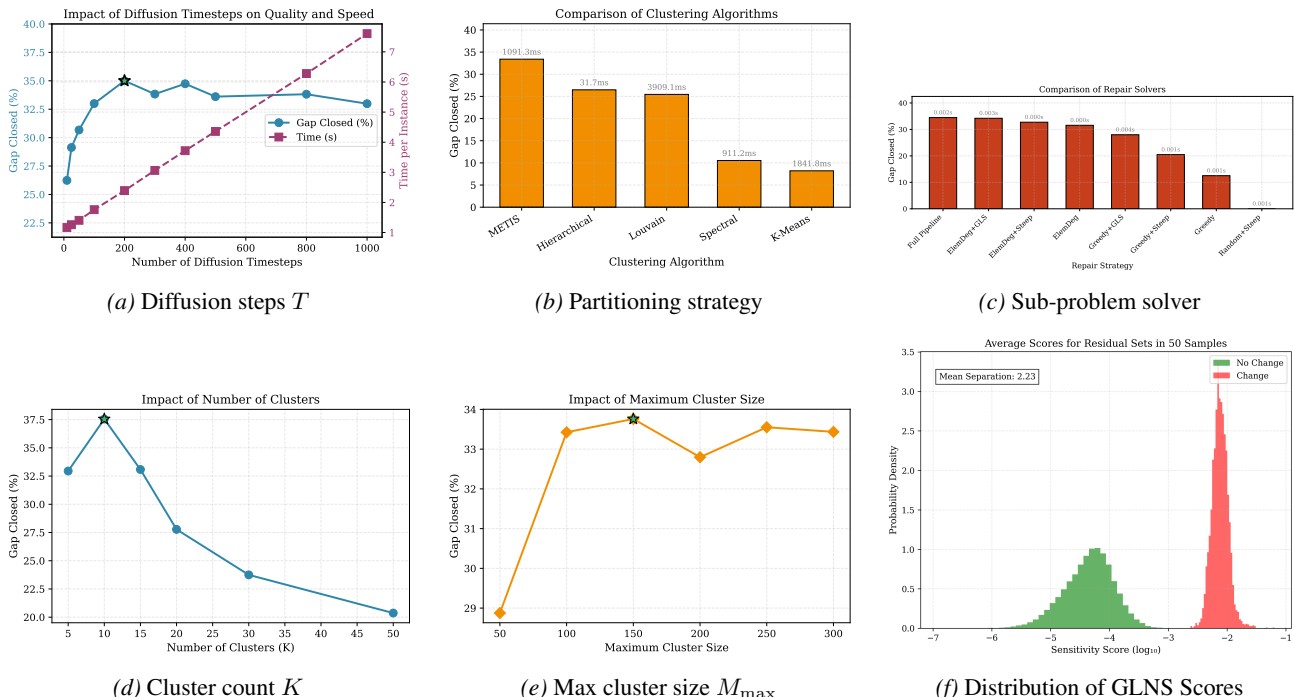

*Figure 7.* Design Ablation and Sensitivity Analysis. We systematically evaluate the impact of framework components on optimality gap closure. (a) Increasing diffusion steps improves solution quality, with returns saturating around $T = 200$. (b) Structural partitioning algorithms (METIS, Louvain) significantly outperform generic clustering (K-Means, Spectral), confirming that preserving topological locality is critical for effective repair. (c) Performance robustness across different sub-problem solvers. (d–e) Stability across a wide range of decomposition granularities (cluster count $K$ and size $M_{\max}$), validating the adaptive decomposition strategy. (f) Bimodal separation of trajectory scores $\mathcal{S}_i$ enables targeted repair.

**Diffusion Horizon and Solver Synergy.** Figure 7a demonstrates that solution quality increases with the diffusion horizon $T$, with returns saturating around $T = 200$. While extended timesteps ($T = 1000$) provide marginal quality gains, they incur a substantial temporal penalty, increasing inference time from 2s to nearly 8s per instance. Analysis of sub-problem solver complementarity (Figures 7c) reveals that while the generative structural prior is effective regardless of the local solver's strength, the combination of trajectory sensitivity with `Steepest Descent` yields the highest absolute performance, nearing 35% gap closure.

**Separability of Trajectory Scores.** The core novelty of our generative signal lies in its ability to distinguish variables involved in bottleneck constraints. Figure 7f visualizes the distribution of log-transformed sensitivity scores $\mathcal{S}_i$ for residual sets. We observe a clear bimodal distribution. This statistical separation indicates that the diffusion trajectory effectively segregates sets involved in "high-tension" constraints (red) from those that should remain fixed (green). This high-fidelity signal aids the robust scale extrapolation observed in Section 5.4, as it allows the model to identify critical repair neighborhoods even when costs are shifted.

### F.3. Sensitivity of LNS Baselines to Wall-Clock Budget

A common concern when comparing learned LNS variants to stochastic destroy–repair baselines is whether the baselines are disadvantaged by the chosen wall-clock budget—for example, whether `Random-LNS` or `ALNS` would close the performance gap given slightly more time per iteration. To make this explicit, Figure 8 reports the primal gap of `Random-LNS` and `ALNS` on the Unicost Rail benchmark as a function of increasing wall-clock budgets (0.1s, 0.5s, and 1.0s), using the same destroy–repair backend and implementation employed throughout the paper.

Both stochastic baselines improve monotonically as the time budget increases, confirming that their performance is time-sensitive and not saturated at the smallest budget. However, the improvement over this range is gradual, and the relative ordering between `Random-LNS` and `ALNS` remains unchanged. This budget sweep therefore serves as a fairness diagnostic: it shows that the baseline curves behave as expected under additional compute, and that the gaps reported in the main

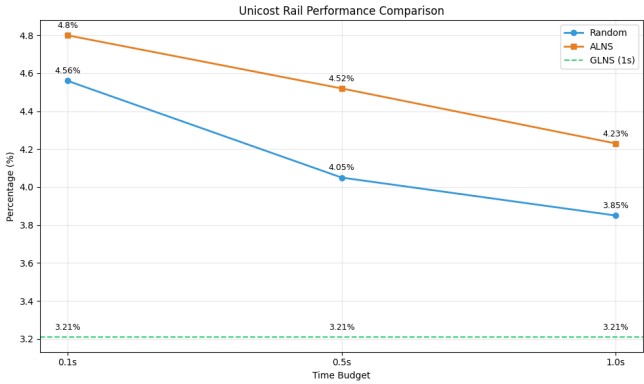

*Figure 8.* Primal gap (%) on the Unicost Rail benchmark as a function of wall-clock budget for stochastic LNS baselines (`Random-LNS` and `ALNS`). The curves illustrate how baseline performance evolves with additional compute. The dashed line indicates the performance of `GLNS` under the same 1-second budget and is shown for reference.

*Table 12.* Paired statistical comparison of GLNS against LNS baselines under the same evaluation protocol (paired per instance). We report mean optimality gap (%) $\pm$ standard deviation, Wilcoxon signed-rank test $p$-values, and paired Cohen's $d$ effect sizes.

| Comparison | GLNS Gap (%) | Baseline Gap (%) | Wilcoxon $p$ | Cohen's $d$ |
|---|---|---|---|---|
| GLNS vs Random-LNS | $3.73 \pm 3.00$ | $4.99 \pm 3.57$ | $< 10^{-4}$ | 1.220 |
| GLNS vs ALNS | $3.73 \pm 3.00$ | $4.77 \pm 3.48$ | $< 10^{-4}$ | 1.066 |

experiments are unlikely to be explained by an overly small or pathological choice of time limit.

Importantly, even under an equal wall-clock budget of 1 second, `GLNS` achieves a lower primal gap than both stochastic baselines. In this sense, the results provide evidence that GLNS exhibits more favorable scaling behavior with respect to time budget than unguided or adaptively weighted stochastic LNS variants.

### F.4. Statistical Significance under an Equal-Iteration Budget

To assess whether GLNS yields consistent improvements over classical LNS baselines beyond aggregate averages, we perform paired statistical tests under an *equal-iteration* evaluation protocol. Specifically, all methods are run for the same number of LNS iterations (where one iteration corresponds to one destroy–repair step followed by the standard local improvement routine used by the method). We then compute the optimality gap (%) of each method with respect to the best known solution and compare GLNS to each baseline using a paired Wilcoxon signed-rank test (two-sided), which does not assume normality. We additionally report paired Cohen's $d$ as an effect-size measure.

Table 12 summarizes the results. Under this equal-iteration protocol, GLNS achieves a lower mean optimality gap than both Random-LNS and ALNS. The differences are statistically significant ($p < 10^{-4}$ in both cases), with large effect sizes (Cohen's $d = 1.220$ vs. Random-LNS and $d = 1.066$ vs. ALNS), indicating that the improvements are both statistically reliable and practically meaningful in terms of solution quality per iteration.

### F.5. Large-Scale Regime Analysis

We further evaluate the methods on an extended benchmark of 200 large-scale set cover instances. The purpose of this experiment is not to study CP-SAT in isolation, but to more precisely characterize the regimes in which GLNS is most effective. In particular, we compare GLNS against exact and mixed-integer optimization solvers under increasing time budgets, where scalability and anytime behavior become critical.

Tables 13 and 14 report the primal integral and final primal gap, respectively. The primal integral measures anytime performance by integrating the primal gap over the full runtime horizon, while the final primal gap measures only the quality of the final incumbent at the end of the allotted time. Thus, the two metrics distinguish methods that rapidly produce strong solutions from methods that may eventually improve but spend a substantial fraction of the budget with weaker incumbents.

The results highlight two complementary regimes. At short time limits, GLNS is highly competitive because it rapidly

| Method | 10s | 20s | 30s | 60s | 90s | 120s |
|---|---|---|---|---|---|---|
| GLNS | **0.223 ± 0.008** | **0.437 ± 0.015** | 0.646 ± 0.022 | 1.234 ± 0.045 | 1.780 ± 0.068 | 2.293 ± 0.091 |
| CP-SAT | 0.232 ± 0.008 | 0.449 ± 0.015 | **0.638 ± 0.023** | **0.919 ± 0.042** | **0.977 ± 0.059** | **1.013 ± 0.073** |
| CP-SAT cold | 2.895 ± 0.067 | 5.789 ± 0.133 | 8.097 ± 0.145 | 14.916 ± 0.191 | 19.148 ± 0.360 | 19.148 ± 0.360 |
| SCIP | 0.233 ± 0.008 | 0.458 ± 0.015 | 0.684 ± 0.023 | 1.360 ± 0.045 | 2.036 ± 0.068 | 2.712 ± 0.091 |

*Table 13.* Large-scale set cover primal integral on an extended benchmark of 200 instances. Lower is better, and bold values indicate the best result for each time limit. CP-SAT COLD denotes the standard CP-SAT solver run from scratch, whereas CP-SAT denotes the same solver initialized with refined feasible solutions. Entries are reported as mean ± standard error of the mean.

| Method | 10s | 20s | 30s | 60s | 90s | 120s |
|---|---|---|---|---|---|---|
| GLNS | **2.18 ± 0.08** | 2.11 ± 0.07 | 2.06 ± 0.07 | 1.88 ± 0.08 | 1.77 ± 0.08 | 1.66 ± 0.08 |
| CP-SAT | 2.28 ± 0.08 | **1.98 ± 0.08** | **1.82 ± 0.08** | **0.26 ± 0.06** | 0.14 ± 0.05 | 0.10 ± 0.05 |
| CP-SAT cold | 28.95 ± 0.67 | 23.08 ± 0.26 | 22.73 ± 0.24 | 14.11 ± 0.77 | **0.00 ± 0.00** | **0.00 ± 0.00** |
| SCIP | 2.25 ± 0.08 | 2.25 ± 0.08 | 2.25 ± 0.08 | 2.25 ± 0.08 | 2.25 ± 0.08 | 2.20 ± 0.08 |

*Table 14.* Large-scale set cover final primal gap on an extended benchmark of 200 instances. Lower is better, and bold values indicate the best result for each time limit. CP-SAT COLD denotes the standard CP-SAT solver run from scratch, whereas CP-SAT denotes the same solver initialized with refined feasible solutions. Entries are reported in percentage points as mean ± standard error of the mean.

constructs good feasible solutions. This is reflected in the primal-integral results, where GLNS obtains the best scores at 10 and 20 seconds and remains close to the best method at 30 seconds. As the time budget increases, initialized CP-SAT can exploit the refined incumbent solutions and further improve the final gap. However, this improvement relies on the availability of strong initial feasible solutions and does not reflect the behavior of CP-SAT from scratch.

The comparison with CP-SAT COLD is therefore primarily a scalability reference point. CP-SAT COLD denotes the standard CP-SAT solver run from scratch, while CP-SAT denotes CP-SAT initialized with refined solutions. On these larger instances, cold CP-SAT is substantially more computationally constrained than GLNS in the early and intermediate regimes. For example, although cold CP-SAT eventually reaches a zero final gap at the longest time limits, its primal integral remains much worse than the other methods, indicating that it spends most of the runtime with poor incumbents. This suggests that, as instance size increases, exact search becomes increasingly limited by the cost of finding and improving feasible solutions, whereas GLNS is designed to produce high-quality incumbents quickly through learned neighborhood construction.

Overall, these results clarify the role of GLNS in the large-scale setting: GLNS is particularly effective in computationally constrained regimes where strong feasible solutions are needed quickly, and its solutions can also serve as useful incumbents for downstream exact optimization. The main advantage is therefore not that GLNS dominates exact solvers at every time horizon, but that it provides strong anytime performance precisely in the regimes where exact solvers become bottlenecked by scale.

## G. GLNS on Other Combinatorial Optimization Problems

To assess the applicability of GLNS beyond the Set Cover problem, we evaluate it on three additional NP-hard combinatorial optimization problems: Maximum Coverage, Set Packing, and Vertex Cover. For each problem, GLNS is trained from scratch using the same architecture and training protocol, and evaluated on a held-out test set. This setting is designed to test whether the GLNS pipeline transfers as a general procedure across problem definitions, rather than to compete with highly specialized, domain-specific heuristics.

**Experimental Setup.** For each problem type, we train the same bipartite GNN and discrete diffusion model from scratch on $N = 2000$ synthetic training instances of fixed size $300 \times 600$, and evaluate on a held-out test set of $N = 200$ instances. We repeat evaluation across multiple independent solver seeds to assess run-to-run stability. The model architecture, training hyperparameters, and inference budget are kept identical across all problems; only the feasibility checker and the greedy repair operator within the LNS loop are adapted to reflect problem-specific constraints.

**Datasets.** Training and test instances are generated using an adversarial synthetic generator with structured "gadgets" designed to expose failure modes of myopic heuristics. Synthetic instances allow precise control over combinatorial structure and difficulty, enabling systematic evaluation of algorithmic behavior that is difficult to isolate using standard benchmark libraries. Specifically:

*Table 15.* Performance across different combinatorial optimization problems. **Gap**: Primal Gap relative to optimal (lower is better). **Imp.**: Improvement over Greedy (higher is better). **Clos.**: Percentage of the Greedy–Optimal gap closed (higher is better). Reported values are means over the full evaluation set; $\pm$ denotes the standard deviation across independent solver seeds, computed on the dataset-level mean for each seed.

| Method | Set Cover | | | Max Coverage | | | Set Packing | | | Vertex Cover | | |
|---|---|---|---|---|---|---|---|---|---|---|---|---|
| | Gap(%) | Imp.(%) | Clos.(%) | Gap(%) | Imp.(%) | Clos.(%) | Gap(%) | Imp.(%) | Clos.(%) | Gap(%) | Imp.(%) | Clos.(%) |
| GLNS | **2.38 ± 0.04** | **18.9 ± 0.11** | **90.8 ± 0.1** | **0.86 ± 0.03** | **1.1 ± 0.08** | **51.7 ± 0.4** | **5.89 ± 0.05** | **12.6 ± 0.1** | **64.1 ± 0.2** | **1.03 ± 0.03** | **1.0 ± 0.06** | **49.9 ± 0.3** |
| GLNS (Final-Only) | 2.45 ± 0.07 | 18.8 ± 0.08 | 90.7 ± 0.2 | 1.45 ± 0.04 | 0.5 ± 0.16 | 22.1 ± 0.5 | 5.90 ± 0.07 | 12.5 ± 0.23 | 64.0 ± 0.3 | 1.49 ± 0.05 | 0.5 ± 0.09 | 25.8 ± 0.7 |
| GNN-LNS | 2.55 ± 0.03 | 18.8 ± 0.13 | 90.2 ± 0.5 | 1.89 ± 0.00 | 0.0 ± 0.00 | 0.0 ± 0.00 | 11.79 ± 0.25 | 5.5 ± 0.22 | 27.9 ± 0.6 | 1.60 ± 0.02 | 0.4 ± 0.07 | 21.2 ± 0.3 |
| Random-LNS | 13.85 ± 0.06 | 9.9 ± 0.11 | 47.8 ± 0.2 | 1.30 ± 0.06 | 0.6 ± 0.09 | 29.8 ± 0.5 | 9.20 ± 0.08 | 8.6 ± 0.3 | 43.7 ± 0.6 | 1.32 ± 0.08 | 0.7 ± 0.11 | 33.8 ± 0.4 |

- *Set Cover & Vertex Cover:* Instances include locally attractive structures (e.g., low-cost sets or low-degree nodes) that fail to cover rare elements or critical edges, requiring coordinated non-local modifications.

- *Maximum Coverage:* Instances contain large decoy sets with substantial overlap, such that effective solutions require identifying complementary subsets under a global budget constraint.

- *Set Packing:* Instances are constructed with interfering modules that require discovering disjoint substructures to avoid constraint violations.

**Results.** Table 15 reports performance on the held-out test sets. We measure the primal gap relative to the optimal solution (**Gap**), the improvement over a greedy baseline (**Imp.**), and the percentage of the greedy–optimal gap closed (**Clos.**). Optimal solutions are obtained using an exact CP-SAT solver with a fixed per-instance time limit. Reported values are means over the test set, with standard deviations computed across independent solver seeds on the dataset-level mean.

Across all four problem domains, GLNS consistently outperforms both the greedy baseline and Random-LNS. The improved performance of GLNS relative to GNN-LNS highlights the benefit of incorporating an iterative generative refinement process, compared to a single-pass supervised prediction of repair sets. Furthermore, consistent with observations in the main text, utilizing the full diffusion trajectory (GLNS) yields consistently higher closure rates than relying solely on the final denoised output (GLNS Final-Only), indicating that intermediate refinement steps contribute meaningfully to solution quality. In particular, for Set Packing and Maximum Coverage, where constraints and objectives differ substantially from Set Cover, GLNS closes 64.1% and 51.7% of the greedy–optimal gap, compared to 43.7% and 29.8% for Random-LNS, respectively. Taken together, these results suggest that diffusion-guided LNS prioritizes structurally related variables for coordinated modification based on graph structure, even when moving between covering, packing, and maximization objectives.

