# OpenReview forum: "Generative Large Neighborhood Search: Scalable Set Cover Optimization via Discrete Diffusion"
_ICML.cc/2026/Conference — ICML 2026 regular_

### Official Review · Reviewer_WRvs · 2026-02-24

**Soundness:** 3
**Presentation:** 2
**Significance:** 2
**Originality:** 3
**Overall Recommendation:** 4
**Confidence:** 3

**Summary:**

This paper proposes Generative Large Neighborhood Search (GLNS) for large-scale Set Cover, reframing neighborhood selection as a generative discrete diffusion problem rather than a pointwise prediction task. GLNS uses halo-based decomposition to extract bounded subproblems and then leverages diffusion denoising trajectory instability to identify structurally coupled variables for coordinated destroy-and-repair.

**Compliance With Llm Reviewing Policy:**

Affirmed.

**Final Justification:**

My main concerns are basically addressed. That said, when the authors discuss the background of this work using a broader notion of “Neural CO,” the paper would still benefit from stronger related-work coverage and discussion, even if empirical comparison is not feasible. Overall, I maintain my score and recommend weak acceptance.

**Key Questions For Authors:**

1. Hyper-parameter studies. How sensitive are results to halo construction choices (e.g., $k$-hop expansion, $N_\max$, and the seed sampling temperature $\tau$) and to the METIS partition granularity $K$?
2. Ablations isolating the "trajectory" contribution. When comparing trajectory-based scoring to "diffusion-final" variants, are diffusion step counts and wall-clock budgets matched exactly (including similarity construction + clustering/partitioning), and does the advantage persist when both are tuned per regime?
3. Wider applicability. The paper suggests potential relevance beyond SCP. Could the authors provide clearer evidence regarding which constraint families or graph structures preserve the connection between *trajectory sensitivity* and *structural coupling*, and in which settings this relationship deteriorates (e.g., denser constraints or different feasibility structures)? In addition, I recommend placing greater emphasis in the main text on experiments involving other CO problems (currently summarized in Table 14), along with more comprehensive comparisons to neural baselines beyond the GLNS variants. This would broaden the scope of the paper and better align it with current mainstream research in the NCO community.

**Note.** Despite the noted weaknesses, I appreciate the overall completeness of this work, particularly the extensive experiments, comprehensive discussions, and detailed disclosure of design choices. Accordingly, I recommend a weak accept at this stage, contingent on the authors’ responses to the concerns raised and their commitment to refining the manuscript upon the suggestions.

**Limitations:**

yes

**Strengths And Weaknesses:**

**Strengths**
1. Using denoising trajectory dynamics (not just final diffusion output) as a heuristic for neighborhood selection (via the sensitivity formulation and affinity modulation) is a clear conceptual contribution.
2. The halo procedure targets constant-bounded neural inference independent of the global instance size, which directly addresses the million-variable regime.
3. The empirical investigations, based on the reported results, are solid and promising, showing consistent gains across cost regimes and competitive performance against strong solvers (e.g., Gurobi/CP-SAT/SCIP).

**Weaknesses & Suggestions**

1. The quality of the figures needs substantial improvement, particularly in terms of resolution (vector graphics are at least expected). The current illustrations also appear to resemble an AI-generated or assisted style, especially in color choices, text layout, and readability. While this is not a critical issue, remaking these visual elements is recommended.
2. The state-change GNN is trained using (near-)optimal labels produced by an exact solver pipeline (SCIP with a time limit), which may be expensive or unavailable in other domains and complicates reproducibility for new datasets.
3. Many important sensitivity experiments are deferred to the appendix. Though extensive, the appendix contains a large amount of information without clear prioritization, and its organization makes it difficult to follow. The authors may consider moving or hghlighting key results, such as those shown in Fig. 8, into the main text, together with clearer discussions on sensitivity and parameter selection principles.
4. Some more recent and advanced diffusion-based baselines are missing. E.g., T2T and Fast-T2T directly enhance DIFUSCO while maintaining the core discrete diffusion pipeline.
5. As a submission to the *discrete_and_combinatorial_optimization* primary area, the coverage of related work on neural combinatorial optimization (NCO), especially the generative CO (GenCO) paradigm (which is the major technique adopted in this paper), appears somewhat limited. Below is a rough collection of representative (but not limited to) works that are worth citing, if not discussed in greater depth or compared empirically. Introducing subsections (e.g., before Appendix B.1) that expand the overview of classical NCO methods and more clearly articulate the landscape of generative CO approaches would be highly beneficial, and at a minimum is necessary for proper positioning within the literature and, by convention, for demonstrating adequate sophistication of the field's current state.

[1] POMO: Policy Optimization with Multiple Optima for Reinforcement Learning, NeurIPS 2020 (classic NCO solver)

[2] MatNet: Matrix Encoding Networks for Neural Combinatorial Optimization, NeurIPS 2021 (classic NCO solver)

[3] Sym-NCO: Leveraging Symmetricity for Neural Combinatorial Optimization, NeurIPS 2022 (classic NCO solver)

[4] DIFUSCO: Graph-based Diffusion Solvers for Combinatorial Optimization, NeurIPS 2023 (classic GenCO method)

[5] T2T: From distribution learning in training to gradient search in testing for combinatorial optimization, NeurIPS 2023 (classic GenCO method)

[6] BQ-NCO: Bisimulation quotienting for efficient neural combinatorial optimization, NeurIPS 2023 (classic NCO solver)

[7] Unsupervised learning for combinatorial optimization needs meta learning, ICLR 2023 (classic NCO solver)

[8] Fast T2T: Optimization Consistency Speeds Up Diffusion-Based Training-to-Testing Solving for Combinatorial Optimization, NeurIPS 2024 (classic GenCO method)

[9] A diffusion model framework for unsupervised neural combinatorial optimization, ICML 2024 (classic GenCO method)

[10] GLOP: Learning Global Partition and Local Construction for Solving Large-scale Routing Problems in Real-time, AAAI 2024 (classic method for generalization/scalability)

[11] UDC: A Unified Neural Divide-and-Conquer Framework for Large-Scale Combinatorial Optimization Problems, NeurIPS 2024 (classic method for generalization/scalability)

[12] COExpander: Adaptive Solution Expansion for Combinatorial Optimization, ICML 2025 (recent GenCO method)

[13] GOAL: A Generalist Combinatorial Optimization Agent Learner, ICLR 2025 (recent NCO solver)

[14] Preference optimization for combinatorial optimization problems, ICML 2025 (recent NCO solver)

[15] UniCO: On Unified Combinatorial Optimization via Problem Reduction to Matrix-Encoded General TSP, ICLR 2025 (recent NCO solver)

[16] BOPO: Neural Combinatorial Optimization via Best-anchored and Objective-guided Preference Optimization, ICML 2025 (recent NCO solver)

[17] DualOpt: A Dual Divide-and-Optimize Algorithm for the Large-scale Traveling Salesman Problem, AAAI 2025 (recent NCO solver)

[18] ML4CO-Bench-101: Benchmark Machine Learning for Classic Combinatorial Problems on Graphs, NeurIPS 2025 (recent NCO benchmark)

[19] UCPO: A Universal Constrained Combinatorial Optimization Method via Preference Optimization, AAAI 2026 (recent NCO solver)

---

> ### Author Rebuttal · Authors · 2026-03-30
>
> We thank the reviewer for their constructive feedback. We address each concern below and hope these clarifications resolve the main issues raised:
> - **Presentation and visibility.** We appreciate these suggestions. In the revision, we will (i) remake the figures with cleaner layout and readability, (ii) expand the related-work discussion as suggested (iii) summarize the most important ablation and sensitivity results in the main text. We also note that the main body already includes a trajectory-vs-final comparison in Table 2 across benchmarks and cost regimes; in the revision, we will make the relation between Table 2 and the stricter matched-control study in Appendix F.1 / Table 11 more explicit.
> - **SCIP supervision cost.** In our setting, SCIP supervision is used only for training instances, where per-instance solve times are typically around 1–2 seconds. Importantly, the diffusion model itself does not require optimal labels. More broadly, since SCP is NP-hard, exact solutions become impractical already at moderate scales; our methodology is designed with this in mind and does not assume that optimal solutions are widely available at large scale. This reliance on supervised labels is also not unique to our approach: constructive neural baselines such as DIFUSCO, T2T, and Fast-T2T likewise require labeled training data. That said, we agree that in domains where even small training instances are too expensive for exact solvers, this supervision strategy becomes a genuine limitation, and we will make this caveat more explicit in the revision.
> - **T2T and Fast-T2T.** We thank the reviewer for highlighting these works, which we will cite in the revision. We evaluated both methods on synthetic SCP instances and found that GLNS performs best overall. We politely refer the reviewer to our response to Reviewer qKSY for the detailed results, and we will include results on the remaining datasets in the final revision.
> - **Related work coverage.** We thank the reviewer for the additional references. We would also like to kindly clarify that the current survey was centered on the regime studied here rather than aiming to exhaustively cover all neural CO settings. Several of the suggested papers address different problem families (e.g. routing), and are therefore more useful for broader contextualization than as direct baselines for the present study. At the same time, central diffusion-based constructive methods such as DIFUSCO are already included in our comparison results. We will revise the related-work section accordingly.
> - **Hyperparameter sensitivity.** Figure 8 provides a broad sensitivity analysis over diffusion timesteps T, partitioning strategy, sub-problem solver, cluster count K, and maximum cluster size M_{\max}, indicating that GLNS is reasonably robust across these choices. Halo-size sensitivity has not been characterized as systematically, since changing the halo size produces larger graphs and would require retraining the neural components for a fair comparison. More broadly, increasing the neighborhood size can in principle strengthen LNS by enabling better repair neighbourhoods and access to more promising candidates.
> - **Trajectory vs. Diffusion-Final Matching.** The comparisons in Tables 2 and 11 are conducted under exactly matched conditions: identical diffusion step counts, wall-clock budgets, similarity construction, clustering, and partitioning procedures—only the neighborhood-selection signal differs. The advantage of trajectory-based selection persists consistently across benchmarks and cost regimes (Table 2). We also note that we did not perform benchmark-specific tuning for either GLNS or the comparison methods. Overall, GLNS appears reasonably robust, although some settings do achieve somewhat better performance than the ones used in the main experiments (larger number of diffusion steps for instance).
> - **Wider applicability.** Appendix G evaluates GLNS on Maximum Coverage, Set Packing, and Vertex Cover using the same architecture, adapting only the feasibility checker and repair operator to each problem. Across all three problem classes, GLNS consistently improves over the corresponding non-neural baselines (Table 14). Moreover, in each case, using the full trajectory outperforms using only the final diffusion step, suggesting that the trajectory signal transfers beyond SCP to other problem formulations. Appendix C further suggests that this signal is most useful when there are multiple plausible repair choices or when variables must be adjusted jointly. It becomes less informative when most decisions are effectively forced and the solution space contains little ambiguity. We view the broader question of how this behaves under substantially different constraint densities or feasibility structures as an important direction for future work. We kindly refer the reviewer to our response to Reviewer 738Q for a complementary discussion of structurally related problem classes.

---

> > ### Author Rebuttal · Reviewer_WRvs · 2026-04-01
> >
> > Thanks for your rebuttal, which has generally addressed my concerns. I therefore maintain my positive recommendation. Regarding the related work, I understand that routing-centered methods may not be directly comparable. I mentioned them only because they may be relevant to the broader notion of "Neural CO" used when you discuss the background.

---

### Official Review · Reviewer_qKSY · 2026-03-11

**Soundness:** 3
**Presentation:** 3
**Significance:** 3
**Originality:** 3
**Overall Recommendation:** 4
**Confidence:** 4

**Summary:**

This paper studies large-scale Set Cover and proposes GLNS, a neural-symbolic framework that uses diffusion trajectory information to guide neighborhood selection in LNS. Instead of relying only on final denoised predictions, the method exploits instability along the denoising trajectory to identify variables that should be repaired jointly. The paper further combines this idea with decomposition machinery to keep inference manageable on large instances, and evaluates the method on several Set Cover benchmarks under both standard and shifted cost regimes.

Overall, I find the paper interesting and reasonably well motivated. The central idea—using trajectory-level generative information as a repair signal for LNS—is novel and empirically promising. At the same time, I think the paper’s current presentation somewhat overstates how conclusively the experiments establish superiority over the strongest relevant baselines, and I would encourage the authors to strengthen the empirical positioning of the work.

**Compliance With Llm Reviewing Policy:**

Affirmed.

**Final Justification:**

The authors’ supplementary experiments and responses have demonstrated to me the clear superiority of the GLNS algorithm for the Scalable Set Cover problem. However, as the scope of problems addressed by the GLNS algorithm is limited and its performance on other CO problems remains unclear, I shall maintain my score. Nevertheless, given that the authors have thoroughly addressed and presented the GLNS algorithm for the Scalable Set Cover problem in this paper, and have resolved the queries I raised in my review comments, I shall increase my confidence rating.

**Key Questions For Authors:**

1. Could the authors include comparisons against stronger recent generative CO baselines such as T2T [1] and FastT2T [2]. Given the paper’s diffusion/generative positioning, these seem like important reference points.
2. The paper standardizes backbone architecture and repair budget across neural baselines. This is reasonable for controlled comparison, but can the authors discuss whether this may disadvantage methods whose strongest reported performance depends on different architectures or solver couplings?
3. Can the authors provide a sharper empirical isolation of the contribution of trajectory sensitivity itself? For example, how much does it improve over final-step diffusion outputs, simpler uncertainty scores, or non-trajectory affinity measures under matched computational budgets? The current mechanistic story is plausible, but still somewhat indirect.
4. The appendix mentions applicability beyond Set Cover, but the main evidence is still heavily SCP-centered. Which parts of GLNS are truly problem-agnostic, and which parts would need substantial redesign for other integer or graph optimization problems?

Reference:

[1] Li Y, Guo J, Wang R, et al. T2t: From distribution learning in training to gradient search in testing for combinatorial optimization[J]. Advances in Neural Information Processing Systems, 2023, 36: 50020-50040.

[2] Li Y, Guo J, Wang R, et al. Fast t2t: Optimization consistency speeds up diffusion-based training-to-testing solving for combinatorial optimization[J]. Advances in Neural Information Processing Systems, 2024, 37: 30179-30206.

**Limitations:**

I think this is an interesting and potentially impactful paper, but the review would be significantly strengthened by a more competitive and better justified baseline suite, especially with respect to recent generative CO methods, and by a more incisive empirical validation of the core mechanism.

More practically, there are several components in the pipeline, and although the supplementary material improves reproducibility, I still found it somewhat difficult to separate the contribution of the trajectory-based idea from the contribution of the overall engineered system. This does not negate the results, but it does make the paper harder to evaluate cleanly.

**Strengths And Weaknesses:**

Strengths:

The main strength is the methodological idea itself. GLNS goes beyond standard prediction-based or final-sample-based neural LNS by explicitly leveraging trajectory dynamics during diffusion, which is a conceptually interesting way to capture ambiguity and coupling among variables. The paper also compares against several categories of baselines, including constructive neural methods, repair heuristics, neural LNS variants, and exact solvers, and the reported results suggest that GLNS is competitive under short-budget settings and under cost distribution shift.

Weaknesses:

The main weakness is that the paper remains largely empirical and heuristic. Although the intuition behind trajectory sensitivity is appealing, there is limited theoretical justification for why this signal should consistently identify high-value repair neighborhoods. The framework is also fairly complex, involving halo extraction, a bipartite GNN, a discrete diffusion model, similarity construction, partitioning, and symbolic repair, which raises reproducibility concerns. Finally, although the paper claims broader relevance, the main paper is still centered on SCP, so the generality of the approach is not yet fully established.

---

> ### Author Rebuttal · Authors · 2026-03-30
>
> We thank the reviewer for their constructive feedback. We address each concern below and hope these clarifications resolve the main issues raised:
>
> - **T2T and Fast-T2T.** We thank the reviewer for pointing out T2T and Fast-T2T. We evaluated both on synthetic SCP instances of varying difficulty at size $300 \times 600$. GLNS achieved the best overall performance. This is aligned with the methodological distinction between the approaches: T2T and Fast-T2T are still constructive extensions of DIFUSCO, whereas GLNS operates in a destroy-repair regime and is designed to improve a feasible incumbent. In the SCP setting considered here, this distinction appears to matter substantially. We will include the full experiments on the remaining datasets in the final revision.
>
> | Method | Mean Gap (%) ↓ |
> |---|---:|
> | GLNS | **3.31** |
> | Fast-T2T | 11.64 |
> | T2T | 11.89 |
> | DIFUSCO | 19.29 |
>
> - **Backbone standardization.** We standardized the backbone architecture and repair budget across neural baselines to keep the comparison controlled and to isolate algorithmic differences rather than differences in model capacity or solver-side tuning. While we cannot fully rule out some interaction with architecture choice or solver coupling, we expect the qualitative conclusion to remain unchanged: GLNS’s advantage over neural LNS baselines appears to stem primarily from its trajectory-guided neighborhood selection mechanism. We kindly refer the reviewer to Figure 8, which shows that GLNS is reasonably robust across a range of design choices, including diffusion horizon, partitioning strategy, sub-problem solver, and decomposition granularity.
>
> - **Trajectory sensitivity isolation.** The role of trajectory sensitivity is directly isolated in Tables 2 and 11, where GLNS is compared against alternative neighborhood-selection signals while holding the rest of the pipeline fixed. Beyond comparing to the final diffusion step, Table 11 also includes entropy of the final map, final-step affinity, ensemble variance across runs, Monte Carlo dropout, and random size-matched clustering; we kindly refer the reviewer to these results.  In these controlled comparisons, the trajectory-based variant performs best. We kindly refer the reviewer to Appendix F.1 for the detailed ablation and discussion.
>
> - **Theoretical Justification for Trajectory Sensitivity.** We agree that a formal characterization of when trajectory-based signals outperform static embeddings remains an open direction, and note this explicitly in the conclusion. Furthermore, Appendix C provides an energy-based interpretation of why prediction instability along the reverse diffusion process can correlate with structural ambiguity in the optimization landscape. Under the view that the learned denoiser approximates the posterior expectation over high-quality solutions, high trajectory sensitivity arises naturally in two settings: variables with high posterior entropy — those participating in multiple competing optima — and variables strongly coupled to such ambiguous decisions. Conversely, structurally essential or dominated variables exhibit stable trajectories. This taxonomy is supported by the controlled synthetic study in Appendix C.6 (Figures 3–5), where trajectory sensitivity shows strong AUROC separation across the planted regimes.
>
> - **Generality Beyond Set Cover.** Appendix G evaluates GLNS on Maximum Coverage, Set Packing, and Vertex Cover. GLNS consistently outperforms both greedy and Random-LNS baselines across all problem types (Table 14), and the full diffusion trajectory outperforms final-step-only variants in each case — suggesting the trajectory signal transfers across problem definitions rather than being specific to Set Cover's structure.
> The core generative mechanism — discrete diffusion, trajectory sensitivity computation, affinity construction, and METIS partitioning — is problem-agnostic. What requires problem-specific adaptation is the graph representation, the halo decomposition strategy, the feasibility checker within the repair loop. In practice, for the problems evaluated in Appendix G, only the feasibility checker and greedy repair operator were adapted; the rest of the pipeline transferred directly. Problems with substantially different constraint structure — such as routing problems with sequential feasibility — would require more significant redesign of the decomposition and repair components, consistent with our response to Reviewer 738Q on TSP.

---

> > ### Author Rebuttal · Reviewer_qKSY · 2026-04-02
> >
> > The author’s answer has resolved all my queries; I will maintain my rating and increase my confidence level.

---

### Official Review · Reviewer_5Cnp · 2026-03-13

**Soundness:** 4
**Presentation:** 4
**Significance:** 4
**Originality:** 4
**Overall Recommendation:** 5
**Confidence:** 4

**Summary:**

This work proposes a generative approach that performs neighborhood selection in large neighborhood search (LNS) using a discrete diffusion model. The key insight is that the iterative denoising trajectory of the diffusion process exposes structural coupling through prediction instability. Compared with previous work that performs point predictions that score variables independently, the proposed approach can capture the coordinated changes across groups of structurally coupled variables better.

The proposed method first decomposes the problem into sub-problems around high-cost regions so that it can scale to large problem instances. Then a GNN network is trained to identify variables likely to differ between the current solution and an improved local
optimum. Then a diffusion process is applied to identify couples of variables that require repairs. Instead of using the final prediction, the authors propose to utilize the trajectory information to capture the target variables.

The authors conduct extensive numerical analyses to confirm the effectiveness (in terms of primal gaps) and efficiency (in terms of primal integrals). The proposed approach has been compared against multiple baseline methods that are highly related. The authors also verify the scalability and the robustness against cost distribution shift.

**Compliance With Llm Reviewing Policy:**

Affirmed.

**Key Questions For Authors:**

1. In Table 1, the authors state that "To assess algorithmic sample efficiency, all LNS variants are evaluated using a fixed budget of iterations". I try to check the appendices but fail to find how many iterations are used and how the number is decided.
2. Similarly, in Table 3 the authors set a 60-second wall-clock budget. I understand that a 60-second budget represents the time constraint scenario. But it might be better to have a PI-time plot up to several minutes so that the audience have a more thorough understanding of the strengths and/or weaknesses of the proposed approach.
3. I have a general question about the GPU acceleration. Do the authors adopt any special tricks to accelerate the GPU computations. My previous experience was that GPU does not provide computation benefit if parallel computation is not applicable (e.g., batched computation).

**Limitations:**

yes

**Strengths And Weaknesses:**

**Strengths** This paper is clearly a good work.
1. The authors have reasonable insights that are well supported by numerical evidence.
2. The paper was written in outstanding quality. Almost every non-trivial insight or design choice is accompanied with detailed discussion. It is very easy to follow.
3. The problem decomposition and processing in a local region is a valid approach that addresses the scalability of neural modules for solving CO problems.
4. The authors conduct numerical analyses in a good manner and compare the proposed approach with mostly related works.

**Weaknesses** Please refer to the question section.

---

> ### Author Rebuttal · Authors · 2026-03-30
>
> We thank the reviewer for their positive comments, and constructive feedback. We address the main questions below:
> - **LNS iteration budget.** All LNS variants are run with a fixed budget of 100 destroy-repair iterations per instance. We kindly refer to Table 10 in the Appendix. This budget reflects a short-horizon deployment setting while keeping comparisons meaningful across methods.
>
> - **Primal integral over extended horizons.** We thank the reviewer for the suggestion. The 60-second PI results in Table 3 were chosen to reflect the tight-budget regime that motivates neural methods for large-scale SCP. We will include a longer-horizon PI plot in the revision for completeness.
>
> - **GPU Acceleration.** GLNS does use batched computation where possible — in particular, diffusion inference is batched across halo subproblems extracted from the same instance, and GNN inference is similarly batched. The dominant runtime at large scales is halo sampling and clustering, which are CPU-bound operations (Figure 6). We do not use any non-standard GPU acceleration tricks beyond standard batching.

---

### Official Review · Reviewer_738Q · 2026-03-15

**Soundness:** 3
**Presentation:** 3
**Significance:** 2
**Originality:** 2
**Overall Recommendation:** 4
**Confidence:** 3

**Summary:**

The paper proposes a gnn+diffusion based neural solver within the large neighborhood search framework for the set cover problem. The method proceeds by decomposing the problem into smaller subproblems (halo decomposition), leverages a bipartite representation of the problem and a gnn to predict which variables should be changed. To predict 'repair groupings', the proposed approach leverages a discrete diffusion model which is pretrained to produce valid affinity matrices (with the help of scores obtained from trajectory sensitivities) and then finetuned with solver guidance. The method achieves strong results and outperforms neural baselines and heuristics.

**Compliance With Llm Reviewing Policy:**

Affirmed.

**Final Justification:**

The authors have addressed my concerns so I have updated my score to a weak accept. The method seems to work reasonably well and it can also be adapted to other problems. The comparisons on that front are more limited so the potential impact in the space is somewhat limited which is why I can't give a higher score. Still, this is a decent empirical contribution that may be interesting to other researchers in this space.

**Key Questions For Authors:**

- The authors argue that the trajectory of the diffusion model provides additional signal that reveals couplings of predictions that could be jointly repaired. Couldn't that be explicitly predicted more directly with a simple model? How does the paper meaningfully differ from recent neural-net based works that also progressively build up a solution like coexpander (see weaknesses and references there)?
- Related to my comments on the potential applicability of the method: what problems could we apply this framework to? Large TSP instances could also exhibit global dependencies: wouldn't they qualify? Or is it also necessary for the feasibility problem to be 'hard' in some sense. Could you make that a bit more precise?

**Limitations:**

Yes

**Strengths And Weaknesses:**

### Strengths

- Strong empirical results. The method performs well both on large scale instances and in the presence of distribution shift.
- The methodology is well explained and the appendix provides several additional ablations and explanations for the methodology.


### Weaknesses
This is primarily an empirical contribution so my scrutiny will be focused on empirical and experimental aspects of the paper.
- I  am a little uncertain regarding the motivation behind focusing on set cover and how it connects to the experiments that the paper has focused on. The paper argues that 'long-range constraint coupling' could be challenging for neural combinatorial optimization methods and that's why problems like set cover remain challenging. I wonder what other problems would you include in this long range constraint coupling category. In my view that position needs to be better developed since it seems that it is used to motivate the direction that this paper takes. Furthermore, there are several successful methods, trained with various approaches ranging from SSL to RL that show strong results on a wide range of problems including vertex cover, dominating set, max coverage, and so on (e.g., [1] and refs therein).  If those problems qualify, the paper should compare to those methods. I am bringing up 1 as a neural method that utilizes similar tools as this paper, since it adaptively develops a solution using a diffusion model.
-  To add to the previous point, it is certainly fine to focus on just one problem and provide a specialized approach for it but that means that the discussion of the problem in the paper needs be more complete. By that I mean for example the background section which should include a more comprehensive breakdown of the best known approaches to set cover and how they vary as the scale of the instances increases. It seems for that for very large scale instances some modification of greedy could work pretty well [2]. Could your approach scale up to those instances and how would it compare in terms of time and performance? Those ones are up to 1.7m sets.
On the other hand, if the paper does not want to focus on the set cover benchmark, then it would make sense to also bring some comparisons with other works on other related problems that contain long range dependencies.
- The paper has conducted several ablations in the appendix but I believe that a distilled summary should be included in the main paper. There are several moving parts to this work so I believe this would help significantly with motivating the architectural choices.
- The paper makes claims like 'Moreover, models trained on small instances often fail to
generalize to larger scales (Fu et al., 2021; Joshi et al., 2020),
motivating hybrid neural–symbolic approaches that guide
classical optimization procedures rather than directly predicting solutions' which I find to be inaccurate on two fronts. The citation for Fu et al., 2021 demonstrates that while the baselines at the time had this generalization issue, the authors were able to train on small scale instances and generalize to 1-2 orders of magnitude larger instances. This is the case for several successful methods from the literature (e.g., [1,3,4])  so I don't think the comment is representative of the field. That's not to say that size generalization should be taken for granted for any neural net based method. It is certainly an important consideration when designing a new method. However, as it can be seen in the literature, it is often achieved in many different ways (RL, SSL,  and SL) so I think that claim should either be retracted or edited to more accurately reflect the current state of the literature.

Overall, the paper seems like a potentially strong empirical contribution to the neural CO literature but I cannot recommend accepting it just yet until some of my comments and questions are resolved. I will gladly reconsider my assessment after the rebuttal.

1. Ma, Jiale, et al. "Coexpander: Adaptive solution expansion for combinatorial optimization." Forty-second International Conference on Machine Learning. 2025.
2. Cormode, Graham, Howard Karloff, and Anthony Wirth. "Set cover algorithms for very large datasets." Proceedings of the 19th ACM international conference on Information and knowledge management. 2010.
3. Tönshoff, Jan, et al. "One model, any CSP: graph neural networks as fast global search heuristics for constraint satisfaction." arXiv preprint arXiv:2208.10227 (2022).
4. Yuan, Hao, et al. "OPTFM: A Scalable Multi-View Graph Transformer for Hierarchical Pre-Training in Combinatorial Optimization." The Thirty-ninth Annual Conference on Neural Information Processing Systems.

---

> ### Author Rebuttal · Authors · 2026-03-30
>
> We thank the reviewer for the constructive feedback and address the main concerns below:
> - **CoExpander.** We thank the reviewer for pointing out CoExpander, which we will cite explicitly. An algorithmic distinction is that CoExpander progressively commits variables (partial → complete), whereas GLNS improves  a feasible solution by unassigning coupled variables and re-optimizing (complete → partial → complete). For SCP, this matters on two fronts. First, any perturbation during construction can violate global covering feasibility in ways that are not locally detectable. Second, early variable commitments distort marginal costs before the full overlap structure is visible — the marginal value of a set depends on what is already covered, making greedy commitment difficult to recover from constructively. Repair-based methods sidestep both by operating within the feasible region throughout. DIFUSCO and DIFFUCO already capture the constructive diffusion-based formulation underlying CoExpander and are included in our core SCP experiments. In contrast, adapting CoExpander is less straightforward: its formulation assumes node- or edge-selection variables on a standard graph, and its authors explicitly identify extension to more complex constrained problems as an open direction (Appendix H.2 of Ma et al., 2025). To further strengthen the baseline suite, we also added FastT2T and T2T (see Response to Reviewer qKSY).
> - **Why Trajectory.** Tables 2 and 11 isolate the effect of the trajectory signal under matched settings. Beyond comparing to the final diffusion step, Table 11 also includes entropy of the final map, final-step affinity, ensemble variance across runs, Monte Carlo dropout, and random size-matched clustering; we kindly refer the reviewer to these results. Appendix C provides intuition in a toy setting why Trajectory Signal is particularly useful. We will surface the ablations more clearly in the final revision.
> - **Large-Scale SCP.** GLNS is agnostic to the initial solution provider; in our experiments, we use an efficient greedy solver that scales to the instance sizes targeted by Cormode et al. The two approaches operate at different levels of the SCP pipeline: Cormode et al. address large-scale feasible construction, while GLNS performs post-improvement of a feasible incumbent. The relevant comparison is therefore not GLNS vs. Cormode et al., but how much GLNS improves its initial construction solution — which is precisely what our experiments measure comparing multiple improvement methods over the same greedy Solver (see GLNS vs. Repair Baselines or Neural Repair Methods) . We will provide further details about the greedy solver upon acceptance.
>  - **SCP Distinction**. We agree that “long-range constraint coupling” is imprecise and will revise it accordingly. More precisely, in single-graph problems such as MIS and MVC, a violated constraint implicates a fixed local neighborhood, so repair candidates are directly identifiable from adjacency. In SCP, by contrast, dropping a set can leave elements uncovered anywhere in the instance, and which sets can repair this depends on global overlap structure not encoded by local adjacency.  This also clarifies which nearby problems are most similar. Maximum Coverage shares the same bipartite substrate but replaces hard full coverage with a budgeted objective, removing the feasibility constraint that drives non-local repair. Hitting Set is the exact dual of SCP and exhibits the same dynamics by the same argument. Appendix G provides empirical evidence for GLNS on some of these problems. More broadly, the question of which structural properties of CO problems make them more or less amenable to neural methods remains largely open, and we view the present analysis as a small step in that direction.
>
> - **On TSP**. Large TSP instances do involve global cost dependencies, but the feasibility structure is different. In metric TSP, a feasible solution is a tour visiting each node exactly once, and standard local improvement moves operate within that feasible set by replacing edges while preserving the tour structure (Rosenkrantz, Stearns, and Lewis (1977) analyze this setting explicitly, including tours that are locally optimal under multi-edge changes).  By contrast, in SCP, removing a single set can immediately uncover elements anywhere in the instance, so a local edit can create a violated feasibility constraint that must be repaired elsewhere. This difference is what we intended to highlight.
> - **Scale Generalization**. We thank the reviewer for this remark and we will revise the claim accordingly. Our intended point is narrower: the inductive biases that have enabled scale generalization in routing and single-graph settings do not have a direct analogue in bipartite covering problems, where both the set and element dimensions grow and the incidence structure changes qualitatively with scale. We will revise the manuscript to reflect this more accurately.

---

> > ### Author Rebuttal · Reviewer_738Q · 2026-04-04
> >
> > I think the authors have for the most part addressed my comments. I still think the paper would benefit significantly from comparing against methods like coexpander in other well known problems that also have challenging constraints. In any case thank you for your rebuttal.

---

### Decision · Program_Chairs · 2026-04-30

**Decision:**

Accept (regular)

**Comment:**

The reasons to accept this paper are:

1. The approach contains components from a number of works that, in total form a novel contribution. The general setting of using generative models within an LNS to solve column generation problems is one that is quite new, and it is an interesting contribution.

2. SCP is simultaneously narrow and general. On the one hand, we could argue just solving SCP is not enough for ICML. But on the other hand, SCP has wide applicability in column generation frameworks, so this simple, almost "toy" problem, actually ends up being rather important for many different real-world applications. Thus, I think the application is sufficient.

3. The experimental results of the paper are convincing. The reviewers feel there could be better OR baselines included in the work, which is one of the key reservations we have about this approach.

In total, it is hard to give a very strong accept recommendation, hence I have put it in the weak accept category, as it is an interesting contribution that provides good performance on an important class of problems.